# Unified Multimodal Model as Auto-Encoder

## Abstract

The pursuit of unified multimodal models (UMMs) has long been hindered by a fundamental schism between multimodal understanding and generation. Current approaches typically disentangle the two and treat them as separate endeavors with disjoint objectives, missing the mutual benefits. We argue that true unification requires more than just merging two tasks. It requires a unified, foundational objective that intrinsically links them. In this paper, we introduce an insightful paradigm through the **Auto-Encoder lens**, *i.e.*, regarding understanding as the encoder (I2T) that compresses images into text, and generation as the decoder (T2I) that reconstructs images from that text. We argue that: *if the encoder truly "understands" the image, its description should capture all essential structure, and if the decoder truly "understands" the text, it should recover that structure faithfully.* Hence, high-fidelity reconstruction serves as a powerful perspective for genuine multimodal unification, evidencing near-lossless, bidirectional information flow between the two processes. To implement this, we propose **UAE**, where we begin by pre-training the decoder with the proposed 700k long-context image-caption pairs to direct it to "understand" the fine-grained and complex semantics from the text, as longer intermediate text, in our Auto-Encoder framework, can preserve more information from the input image for reconstruction. We then propose **Unified-GRPO** via reinforcement learning (RL) to unify the two, which covers two complementary stages: (1) *Generation for Understanding*, where the encoder is trained to generate informative captions that maximize the decoder's reconstruction quality, enhancing its visual perception; (2) *Understanding for Generation*, where the decoder is refined to reconstruct from these captions, forcing it to leverage every detail and improving its long-context instruction following and generation fidelity. Our empirical results suggest that understanding can largely enhance generation (verified on GenEval), while generation, in turn, notably strengthens fine-grained visual perception like small object and color recognition (verified on MMT-Bench). This bidirectional improvement reveals a deep synergy: under the unified reconstruction objective, generation and understanding can mutually benefit each other, moving closer to truly unified multimodal intelligence.

## 1 Introduction and Motivation

> *"Imagine opening your eyes to a scene, then closing them—your unified brain instantly recalls it, much like an auto-encoder."*

Unifying multimodal models (UMMs) that support both generation and understanding has recently gained increasing popularity in both academia and industry (Wang et al., 2024b; Chen et al., 2025b; Wu et al., 2025a; Xie et al., 2024b; Pan et al., 2025; Gupta et al., 2022; Zhou et al., 2024; Yan et al., 2025; Deng et al., 2024; Ge et al., 2024). However, directly unifying the understanding and generation models can lead to a sub-optimal result, as most existing arts on UMMs (Wu et al., 2025a; Pan et al., 2025; Chen et al., 2025a) suggest that optimizing diffusion-based generative objectives negatively degrade the understanding capability and learned representations (and conversely), making joint training brittle.

Consequently, some existing works decouple the UMM problem (Wu et al., 2025a; Qu et al., 2025), training understanding and generation modules separately, and missing out on potential cross-task mutual benefits. These design choices and empirical observations have dampened confidence in truly unified systems: absent demonstrable mutual gains, "unification" collapses into training two

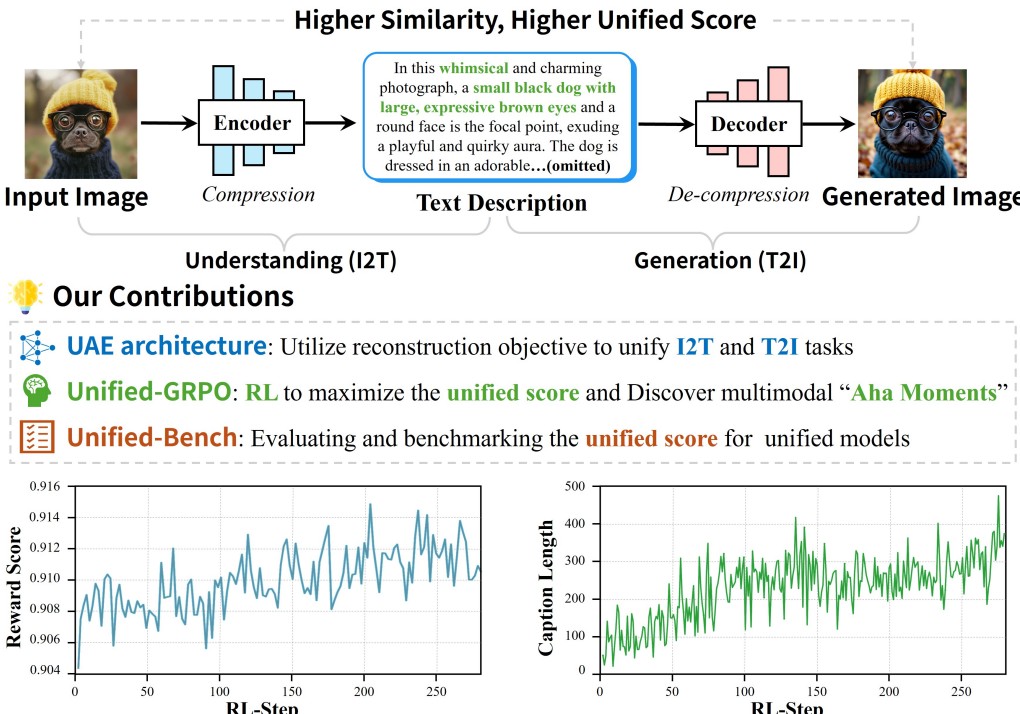

Figure 1: Illustration of the key insight of our **UAE**, an Auto-Encoder inspired design, for unified multimodal understanding and generation. We treat the understanding model as the encoder and the generation model as the decoder. Using the reconstruction similarity as the unified score, we use RL to maximize it (Unified-GRPO) and utilize it to evaluate the degree of unification (Unified-Bench).

large components side by side. This raises an important question: *Can understanding truly benefit generation, and vice versa?* And how can this interplay be optimized in a complementary, mutually reinforcing way, a direction still unexplored? A genuinely unified approach, however, should deliver explicit, bidirectional gains, leveraging each task to strengthen the other, rather than merely bridging them as independent parts.

In this paper, we argue that the **Auto-Encoder view is central to unifying multimodal understanding and generation.** In this view (Fig. 1), the two tasks are symmetric and complementary: the encoder compresses visual content into a descriptive, compact caption (I2T), and the decoder reconstructs it to pixels (T2I). We leverage the similarity between the input image and reconstruction image as the key objective to optimize both tasks, where successful reconstruction (higher similarity) indicates a more coherent bidirectional information flow between the understanding (encoding) and generation (decoding), leading to a higher **unified score** that indicates a more unified system between the understanding and generation.

To operationalize this, we introduce **UAE**, a new framework for unified multimodal learning, as illustrated in Fig. 2. Since reconstructing an image from input to output requires an ultra-detailed textual caption that maximally preserves the original image's information, we propose **LongCap-700k**, a highly descriptive dataset for text-to-image generation, designed to train the decoder to "understand" long-context inputs. We then pre-train the decoder on these long-context image captions (at 1024 resolution) to optimize the model for capturing fine-grained visual semantics and compositional structure.

Then, we propose a post-training strategy for UMMs, namely **Unified-GRPO**, the first reinforcement learning (RL) approach to benefit both understanding and generation modules with a unified objective. Our Unified-GRPO covers two complementary stages: *(1) Generation for Understanding*: the encoder is trained to produce the highly descriptive, generation-friendly captions that maximize the decoder's reconstruction quality, thereby strengthening visual perception; *(2) Understanding for Generation*: the decoder is refined to reconstruct from the text, forcing it to leverage every detail and improving long-context instruction following and generation fidelity.

Figure 2: The overall workflow of our **UAE**, consisting of three stages: long-context pre-training (stage-1), generation for understanding (stage-2), and understanding for generation (stage-3). We name our post-training method as Unified-GRPO (the last two RL stages), which utilize a single, unified reconstruction objective for optimization.

During experiments, *we observe an emergent "aha moment" in multimodal learning*: as RL progresses, the encoder autonomously generates longer, richer captions, while the decoder concurrently improves its ability to interpret them, yielding reconstructions of striking fidelity, as demonstrated in Fig. 1. This co-evolution offers compelling evidence of progress toward genuine multimodal unification. Our empirical results support that the understanding can greatly improve the generation performance (e.g., from 0.73→0.86 on GenEval (Ghosh et al., 2023) and 0.296→0.475 on GenEval++ (Ye et al., 2025)), while generation also notably enhances specifically dimensions of the understanding, particularly fine-grained visual recognition and perception, e.g., from 0.05→0.45 on small object detection and from 0.15→0.75 on Person ReID of the MMT-Bench (Ying et al., 2024), consistent with the findings reported by Ross (Wang et al., 2024a)). These results demonstrate that understanding and generation can indeed mutually benefit each other to a certain extent by training with a unified reconstruction objective.

In summary, our work makes the following contributions:

- We propose **UAE**, **the first work based on an Auto-Encoder principle for the unified multimodal model**, casting understanding as the encoder (I2T) and generation as the decoder (T2I), with reconstruction as a measurable signal of cross-modal information coherence. This resolves the long-standing schism between understanding and generation and provides an actionable, verifiable objective for unified multimodal models (UMMs).
- We develop *Unified-GRPO*, the **first RL-based post-training method to achieve the mutual bonus** that improves the unification between generation and understanding. This bidirectional optimization forms a positive feedback loop toward genuine unification.
- Our empirical results demonstrate that the understanding and generation models can indeed **mutually benefit via a unified reconstruction objective**.
- We release *Unified-Bench*, the tailored benchmark explicitly designed to **measure the degree of unification** in UMMs, rather than individually evaluating the generation or understanding capabilities.
- We provide *LongCap-700k*, **a highly descriptive image caption dataset** for text-to-image generation, enabling the generation model to "understand" rich, detailed textual descriptions, fine-grained semantics and their relationship. Within our auto-encoder framework, the intermediate text representation is inherently long to preserve maximal visual information.

## 2 UAE METHODOLOGY

### 2.1 ARCHITECTURE

**Overview.** Our system follows a compact *encode–project–decode AE-based* design, the most simple and intuitive way to implement our "Auto-Encoder framework", which couples a Large Vision–Language Model (LVLM) for multimodal understanding with a strong diffusion transformer (DiT) for image synthesis. The LVLM converts the input (image and optional prompt) into a rich semantic representation; a lightweight projector then maps this representation to the decoder's conditioning space; finally, the diffusion model expands this condition into pixels. This separation keeps the interface minimal, preserves the strengths of each component, and makes the system modular and scalable. We show the details of the encoder and decoder used in our paper below. More detailed description of the dataset can be seen in Appendix Sec. B.

**Encoder.** We adopt **Qwen-2.5-VL 3B** (Bai et al., 2025) as the base LVLM encoder. It consists of a visual encoder paired with an autoregressive language model capable of processing vision–language inputs. For generation, the LVLM autoregressively processes the prompt and multimodal context to produce a high-dimensional, context-rich representation. Rather than passing raw text to the decoder, we extract the *last hidden state* from the LLM and feed it to a small MLP projector. The projected embedding serves as the decoder's conditioning signal, providing a compact semantic summary grounded in the LVLM's learned world knowledge. **Decoder.** For the visual decoder, we use a well-pretrained diffusion model to reconstruct image pixels from the LVLM's semantic representation. Concretely, we employ **SD3.5-large** (Esser et al., 2024) and add a minimal projector head (two linear layers) to match the LVLM embedding dimension to the conditioning channels expected by SD3.5. During synthesis, the diffusion decoder takes the projected semantic condition as input and then generates images. The detailed architecture can be seen in Appendix Sec. C.

### 2.2 PRE-TRAINING

**Stage-1: Long-Context Pretraining.**

The initial pre-training stage aims to align a DiT decoder with a frozen LVLM encoder. Our training objective is based on a Rectified Flow (RF) formulation (Lipman et al., 2022), which operates within the latent space of a pre-trained VAE. The image $x$ is first encoded into a latent representation $z_1 = \mathcal{E}(x)$. We define a linear path between a standard Gaussian noise vector $z_0$ and the target latent $z_1$ as $z_t = (1 - t)z_0 + t z_1$ for $t \in [0, 1]$. The DiT, framed as a conditional velocity predictor $v_\theta$, is trained to estimate the constant velocity vector of this path, $z_1 - z_0$. The parameters $\theta$ are optimized by minimizing the mean squared error between the predicted and target vectors:

$$\mathcal{L}(\theta) = \mathbb{E}_{z_1 \sim \mathcal{E}(x),\ z_0 \sim \mathcal{N}(0, I),\ t \sim U[0,1]} \left[ \left\| v_\theta(z_t, t, c) - (z_1 - z_0) \right\|^2 \right]. \tag{1}$$

This process trains the DiT decoder to be semantically aligned with the LVLM's descriptive captions, providing a robust foundation for subsequent post-training RL.

### 2.3 POST-TRAINING: UNIFIED-GRPO

**Preliminary of Group Relative Policy Optimization (GRPO).** In GRPO Shao et al. (2024), for a given input, a policy model $\pi$ generates a set of $G$ trajectories, denoted $\{o_i\}_{i=1}^{G}$. The estimation of an advantage $\tilde{A}_i$ for each trajectory is $\tilde{A}_i = \frac{R_i - \text{mean}(\{R_k\}_{k=1}^{G})}{\text{std}(\{R_k\}_{k=1}^{G})}$. The policy's parameters $\theta$ are then updated by maximizing the GRPO objective function:

$$\mathcal{J}(\theta) = \mathbb{E}_{c \sim \mathcal{C},\ \{o_i\}_{i=1}^{G} \sim \pi_{\theta_{\text{old}}}(\cdot | c)}$$
$$\left[ \frac{1}{G} \sum_{i=1}^{G} \frac{1}{T_i} \sum_{t=0}^{T_i - 1} \left( \min\left(r_t^i(\theta)\, \tilde{A}_i,\ \text{clip}(r_t^i(\theta), 1 - \varepsilon, 1 + \varepsilon)\, \tilde{A}_i\right) - \beta\, \mathbb{D}_{\text{KL}}(\pi_\theta \,\|\, \pi_{\theta_{\text{ref}}}) \right) \right], \tag{2}$$

where $r_t^i(\theta)$ is the probability ratio between the current and old policies, and $T_i$ is the length of $i$-th trajectory. In Stage-2 and 3, we introduce the specific instantiations of the policy and the trajectory.

**Stage-2: Generation for Understanding.** In this stage, the LVLM $\pi_\phi$ serves as the policy, while the DiT $p_\theta$ is *frozen* and functions as part of the reward evaluation environment alongside the CLIP

encoder. For each input image $x$, we sample a group of $G$ caption sequences $\{y^{(i)}\}_{i=1}^{G}$ from the old policy $\pi_{\phi_{\text{old}}}(\cdot \mid x)$. From each sequence $y^{(i)}$, we extract the last hidden state $h_T^{(i)}$ to form a condition $c^{(i)} = g(h_T^{(i)})$, which is subsequently used to synthesize an image $\tilde{x}^{(i)} \sim p_{\theta}(\cdot \mid c^{(i)})$. The LVLM's parameters $\phi$ are then updated by maximizing the GRPO objective in Equation equation 2. In this context, each trajectory $o_i$ corresponds to a sampled caption sequence $y^{(i)} = (y_1^{(i)}, \ldots, y_{T^{(i)}}^{(i)})$. The probability ratio is thus defined as $r_t^i(\phi) = \frac{\pi_{\phi}(y_t^{(i)}|x, y_{<t}^{(i)})}{\pi_{\phi_{\text{old}}}(y_t^{(i)}|x, y_{<t}^{(i)})}$. This stage trains the LVLM to emit last-hidden representations that *maximize* the decoder's reconstruction quality.

**Stage-3: Understanding for Generation.** The roles are now reversed: the image generation model $p_{\theta}$ (e.g., DiT) acts as *policy*, while the LVLM is *frozen*, serving to provide conditions $c = \{c_{\text{text}}, c_{\text{img}}\}$ for generation. Note that $c_{\text{img}}$ is an alternative option, as we find that it produces very similar results to those using only the LVLM output caption. We optimize $p_{\theta}$ using the GRPO by sampling reverse-time generation trajectories. For a given condition $c$, the policy $p_{\theta}$ generates a group of $G$ images $\{x_0^i\}_{i=1}^{G}$. In this context, each trajectory $o_i$ corresponds to a full reverse-time sequence $(x_T^i, x_{T-1}^i, \ldots, x_0^i)$, representing the denoising process from an initial noise sample $x_T^i$ to the final image $x_0^i$. The parameters $\theta$ of the generation model are then updated by maximizing the GRPO objective in Equation equation 2. For this stage, the per-step likelihood ratio is given by:

$$r_t^i(\theta) = \frac{p_{\theta}(x_{T-1}^i \mid x_T^i, c)}{p_{\theta_{\text{old}}}(x_{T-1}^i \mid x_T^i, c)}. \tag{3}$$

The stochasticity arises from the SDE sampling of the reverse process.

## 3 *Unified-Bench*: A BENCHMARK TAILORED FOR EVALUATING THE UNIFIED MODELS

**Motivation.** As illustrated in Fig. 1, we view *understanding* (I→T) and *generation* (T→I) as a closed loop whose two halves should *mutually enhance* each other. Judging image realism alone or caption fidelity alone cannot reveal whether a system is truly *unified*. We therefore adopt a reconstruction-based similarity, our **unified-score**, to directly test whether the semantics distilled during understanding are sufficient for faithful regeneration, and whether regeneration in turn validates the completeness of the understanding.

*Protocol-1*: **Evaluation of the unified score from the reconstruction similarity.** To quantify the unified score, we start from 100 diverse source images. The prompt, used to allow the model to generate cpation, is detailed in appendix, Sec. C. The same model then synthesizes an image from its *own* caption. We compute unified scores between the reconstruction and the source using four widely adopted vision backbones, CLIP (Radford et al., 2021), LongCLIP (Zhang et al., 2024), DINO-v2 (Oquab et al., 2023), and DINO-v3 (Siméoni et al., 2025), and report per-backbone similarities and an overall summary.

*Protocol-2*: **Quality Evaluation of the model's *output caption* for reconstruction.** We further evaluate caption quality through pairwise comparisons against various baselines, using four commercial LLM judges: Claude-4.1, GPT-4o, Grok-4, and o4-mini. The prompting strategy is detailed in Appendix Sec. E. For evaluation, we use pairwise winning rate (%), the percentage of times our model is preferred over baselines as the main metric.

## 4 RESULTS

### 4.1 UNIFIED EVALUATION

We assess the unified degree with the proposed Unified-Bench. Tab. 1 shows that our **UAE** achieves the best **Overall** unified score (86.09), surpassing GPT-4o-Image (85.95). Specifically, UAE obtains the top results on CLIP (90.50), DINO-v2 (81.98), and DINO-v3 (77.54), and statistical parity on LongCLIP (94.35 vs. 94.37). These consistent gains across contrastive (CLIP-family) and self-supervised (DINO-family) features suggest that our UAE framework can preserve layout- and texture-level semantics that translate into more faithful reconstructions.

Table 1: **Protocol-1 of Unified-Bench**: comparing of unified score of different methods on Unified-Bench, the tailored benchmark for evaluating the unification between understanding and generation models in the UMMs. **Bold** indicates the best result, and underlined denotes the second best.

| Method | CLIP | LongCLIP | DINO-v2 | DINO-v3 | Overall |
|---|---|---|---|---|---|
| GPT-4o-Image (OpenAI, 2025) | 90.42 | **94.37** | 81.74 | 77.27 | 85.95 |
| BAGEL (Deng et al., 2025) | 88.97 | 93.35 | 78.55 | 73.05 | 83.48 |
| BLIP-3o (Chen et al., 2025a) | 84.84 | 90.24 | 68.31 | 62.86 | 76.56 |
| Janus-Pro (Chen et al., 2025b) | 88.72 | 93.45 | 78.30 | 70.61 | 82.77 |
| OmniGen2 (Wu et al., 2025b) | 88.36 | 93.11 | 77.70 | 74.07 | 83.31 |
| Show-o (Xie et al., 2024a) | 80.18 | 86.75 | 58.20 | 51.51 | 69.16 |
| UniWorld-V1 (Lin et al., 2025) | 85.49 | 91.53 | 72.12 | 66.83 | 78.99 |
| **UAE** | **90.50** | 94.35 | **81.98** | **77.54** | **86.09** |

Table 2: **Benchmarking results of text-to-image generation capability**. We compare our method with other unified multimodal models on GenEval (Ghosh et al., 2024) benchmark. '†' refers to the methods using LLM rewriter. **Bold** indicates the best result, and underlined denotes the second best.

| Method | Single object | Two object | Counting | Colors | Position | Color attribution | Overall |
|---|---|---|---|---|---|---|---|
| Janus Pro (Chen et al., 2025b) | 0.99 | 0.89 | 0.59 | 0.90 | **0.79** | 0.66 | 0.80 |
| MetaQuery-XL[†] (Pan et al., 2025) | - | - | - | - | - | - | 0.80 |
| BLIP3-o 8B (Chen et al., 2025a) | - | - | - | - | - | - | 0.84 |
| UniWorld-V1 (Lin et al., 2025) | 0.99 | 0.93 | 0.79 | 0.89 | 0.49 | 0.70 | 0.80 |
| UniWorld-V1[†] (Lin et al., 2025) | 0.98 | 0.93 | 0.81 | 0.89 | 0.74 | 0.71 | 0.84 |
| OmniGen2 (Wu et al., 2025b) | **1.00** | 0.95 | 0.64 | 0.88 | 0.55 | 0.76 | 0.80 |
| BAGEL (Deng et al., 2025) | 0.99 | 0.94 | 0.81 | 0.88 | 0.64 | 0.63 | 0.82 |
| BAGEL[†] (Deng et al., 2025) | 0.98 | 0.95 | **0.84** | **0.95** | 0.78 | 0.77 | 0.88 |
| **UAE** | **1.00** | 0.89 | **0.84** | 0.90 | 0.71 | 0.79 | 0.86 |
| **UAE**[†] | **1.00** | **0.97** | 0.82 | **0.95** | 0.73 | **0.84** | **0.89** |

**Baseline** **w/ Ours** **Baseline** **w/ Ours** **Baseline** **w/ Ours**

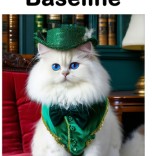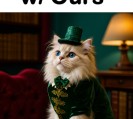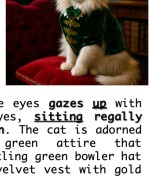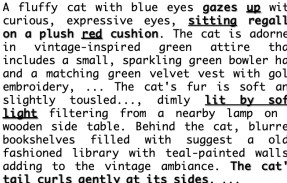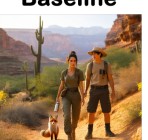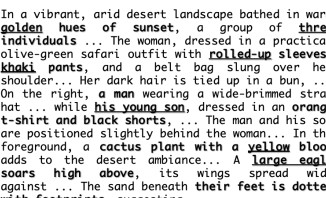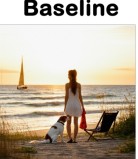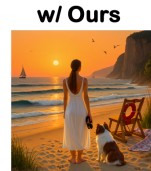

A fluffy cat with blue eyes **gazes up** with curious, expressive eyes, **sitting regally on a plush red cushion**. The cat is adorned in vintage-inspired green attire that includes a small, sparkling green bowler hat and a matching green velvet vest with gold embroidery, ... The cat's fur is soft and slightly tousled..., dimly **lit by soft light** filtering from a nearby lamp on a wooden side table. Behind the cat, blurred bookshelves filled with suggest a old-fashioned library with teal-painted walls, adding to the vintage ambiance. **The cat's tail curls gently at its sides**, ...

In a vibrant, arid desert landscape bathed in warm, **golden hues of sunset**, a group of **three individuals** ... The woman, dressed in a practical olive-green safari outfit with **rolled-up sleeves**, **khaki pants**, and a belt bag slung over her shoulder... Her dark hair is tied up in a bun, ... On the right, **a man** wearing a wide-brimmed straw hat ... while **his young son**, dressed in an **orange t-shirt and black shorts**, ... The man and his son are positioned slightly behind the woman... In the foreground, **a cactus plant with a yellow bloom** adds to the desert ambiance... A **large eagle soars high above**, its wings spread wide against ... The sand beneath **their feet is dotted with footprints**, suggesting...

In this serene, **sunset-hued beach** scene, a woman stands with her back to the viewer, gazing out at the ocean. She has... She carries a pair of **black flip-flops in her right hand**. Her light brown and white dog sits attentively beside her on the sandy shore... To the left, a sailboat sails across the calm sea, its silhouette silhouetted against the warm sky. A wooden **lifeguard chair with a red life buoy** stands near the center-right of the scene, next to a blanket with a **floral pattern** draped over its legs. The beach is **dotted with footprints**, and tall grasses and shrubs frame the scene. **A couple of seagulls fly low in the orange sky**, adding to the tranquil atmosphere...

Figure 3: **Qualitative results on the complex and long-context generation.** Our method can recover very detailed semantics from the highly descriptive input caption over the baseline, demonstrating that improved understanding can notably benefit generation.

## 4.2 MULTIMODAL GENERATION EVALUATION

We evaluate UAE on two standard benchmarks: GenEval and its improved version GenEval++, which probe compositional understanding and instruction-following in increasingly challenging settings. More text-to-image evaluations are in Appendix Sec. E.

**GenEval.** As shown in Tab. 2, without considering LLM rewriting, our UAE attains the best *Overall* score among unified models (**0.86**). It leads on *Counting* (0.84) and *Color attribution* (0.79; **+16** points vs. Bagel's 0.63 and +3 vs. OmniGen2's 0.76), co-leads on *Colors* (0.90), is second-best on *Position* (0.71), and reaches 0.89 on *Two object* (below the strongest 0.94–0.95). When considering LLM rewriting, e.g., using the same rewritten prompts with Bagel, our UAE achieves an overall score of 0.89 on average, demonstrating the SOTA performance in the image generation task.

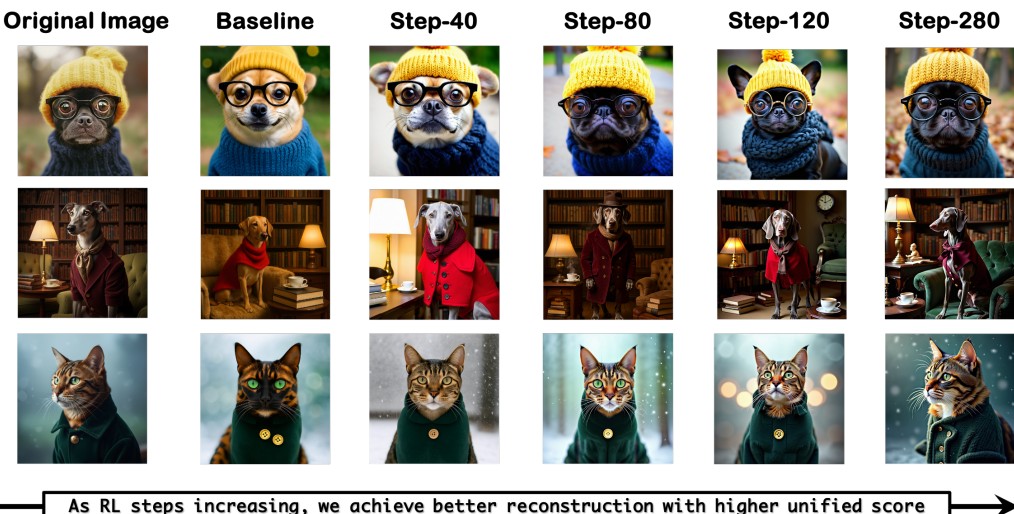

| Original Image | Baseline | Step-40 | Step-80 | Step-120 | Step-280 |

As RL steps increasing, we achieve better reconstruction with higher unified score

Figure 4: **Reconstruction results *vs*. RL training steps.** With the RL steps increasing, the understanding model (encoder) achieves better caption capability to produce a longer, detailed, yet accurate caption to reconstruct the original image comprehensively; while the generation model (decoder) can take the detailed caption as input for better generation. See appendix for more examples.

Table 3: Comparisons of **challenging instruction following generation ability** with other unified multimodal models on Geneval++ (Ghosh et al., 2024). **Bold** indicates the best result, and underlined denotes the second best.

| Method | Color | Count | Color/Count | Color/Pos | Pos/Count | Pos/Size | Multi-Count | Overall |
|---|---|---|---|---|---|---|---|---|
| Janus-Pro (Chen et al., 2025b) | 0.450 | 0.300 | 0.125 | 0.300 | 0.075 | 0.350 | 0.125 | 0.246 |
| T2I-R1 (Jiang et al., 2025) | **0.675** | 0.325 | 0.200 | 0.350 | 0.075 | 0.250 | 0.300 | 0.311 |
| BLIP3-o 4B (Chen et al., 2025a) | 0.125 | 0.225 | 0.100 | 0.450 | 0.125 | 0.550 | 0.225 | 0.257 |
| BLIP3-o 8B (Chen et al., 2025a) | 0.250 | 0.250 | 0.125 | **0.600** | 0.125 | **0.575** | 0.225 | 0.307 |
| OmniGen2 (Wu et al., 2025b) | 0.550 | 0.425 | 0.200 | 0.275 | 0.125 | 0.250 | **0.450** | 0.325 |
| Bagel (Deng et al., 2025) | 0.325 | 0.600 | 0.250 | 0.325 | 0.250 | 0.475 | 0.375 | 0.371 |
| **UAE** | 0.550 | 0.525 | **0.550** | 0.550 | **0.450** | 0.400 | 0.400 | **0.475** |

Table 4: **Protocol-2 of Unified-Bench**: evaluating the quality of output caption of our trained understanding model (3B) against different opponents on Unified-Bench, evaluated by four judge models (using official commercial API). We use the metric of **Pairwise winning rate (%)** for evaluation. The **Avg** column reports the mean score across judges.

| Opponent | # Param | Our Wining Rate (%) | | | | |
|---|---|---|---|---|---|---|
| | | Claude-4.1 | GPT-4o | Grok-4 | o4-mini | Avg |
| GPT-4o (OpenAI, 2025) | - | 47.4 | 89.4 | 30.6 | 21.2 | 47.2 |
| Bagel (Deng et al., 2025) | 7B | 57.7 | 92.9 | 58.3 | 48.2 | 64.3 |
| OmniGen2 (Wu et al., 2025b) | 3B | 67.9 | 97.6 | 63.5 | 56.5 | 71.4 |
| Show-o (Xie et al., 2024a) | 1.3B | 97.8 | 100.0 | 89.8 | 91.0 | 94.7 |
| Qwen-2.5-VL-3B (Bai et al., 2025) | 3B | 76.3 | 99.0 | 67.0 | 63.0 | 76.3 |
| Qwen-2.5-VL-7B (Bai et al., 2025) | 7B | 68.8 | 99.0 | 62.0 | 56.0 | 71.5 |

**GenEval++ (harder compositional control).** GenEval++ (Ye et al., 2025) extends GenEval to prompts with *three or more* objects, each bearing distinct attributes and spatial relations, demanding comprehensive, multi-constraint satisfaction. In Tab. 3, UAE achieves the best *Overall* score (**0.475**), leading on *Color/Count* (0.550) and *Pos/Count* (0.450), with runner-up performance on *Color/Pos* (0.550) and *Multi-Count* (0.400). Qualitative visualizations in Fig. 3 further show accurate attribute binding, disambiguation across multiple entities, and robust position–count consistency under long, constraint-heavy prompts.

Table 5: **Evaluating how "friendly" the output caption is for image generation**. We use the data from Unified-Bench to assess the quality of the captions produced by the understanding model for better text-to-image generation. **Bold** indicates the best result.

| Method | CLIP | LongCLIP | DINO-v2 | DINO-v3 | Overall |
|---|---|---|---|---|---|
| Qwen-2.5-VL-3B (Bai et al., 2025) | 88.34 | 92.62 | 73.91 | 70.02 | 80.85 |
| Qwen-2.5-VL-7B (Bai et al., 2025) | 88.26 | 92.89 | 76.12 | 70.96 | 81.92 |
| **UAE** | **90.50** | **94.35** | **81.98** | **77.54** | **86.09** |

Table 6: **High-level meta-tasks evaluation results on the comprehensive multimodal understanding benchmark:** MMT-Bench (Ying et al., 2024). Accuracy is the metric, and the Overall score is computed as the mean of all displayed subtasks.

| Model | Overall | VR | Loc | Count | HLN | VC | VG | AR | PLP | I2IT | RR | Emo | VI | OCR | DU | IR | 3D |
|---|---|---|---|---|---|---|---|---|---|---|---|---|---|---|---|---|---|
| Frequency Guess | 32.3 | 30.0 | 28.2 | 28.2 | 43.4 | 28.2 | 29.1 | 30.0 | 29.4 | 30.8 | 33.5 | 30.1 | 52.1 | 30.4 | 37.6 | 29.9 | 26.5 |
| Random Guess | 27.9 | 27.1 | 28.1 | 25.0 | 41.6 | 25.0 | 24.8 | 26.6 | 21.2 | 33.4 | 10.5 | 25.4 | 50.8 | 27.2 | 30.3 | 24.3 | 25.5 |
| InternVL-Chat-v1.2-34B | 58.7 | 81.3 | 59.4 | 66.4 | 82.4 | 82.3 | 49.4 | 52.6 | 37.4 | 32.8 | 55.0 | 48.7 | 61.5 | 60.5 | 68.3 | 56.3 | 45.5 |
| Qwen-VL-Plus | 56.8 | 82.6 | 55.3 | 61.1 | 69.9 | 86.5 | 43.6 | 53.4 | 43.1 | 37.8 | 53.0 | 41.6 | 50.3 | 65.6 | 77.3 | 40.7 | 46.5 |
| GPT-4V | 54.1 | 85.3 | 55.6 | 51.6 | 69.6 | 80.3 | 25.0 | 47.7 | 48.2 | 31.8 | 52.5 | 45.1 | 47.9 | 68.0 | 69.8 | 44.9 | 42.0 |
| GeminiProVision | 56.2 | 84.7 | 43.6 | 56.4 | 65.9 | 80.1 | 33.0 | 57.4 | 40.3 | 31.5 | 58.5 | 55.2 | 47.5 | 59.5 | 71.6 | 68.4 | 45.2 |
| DeepSeek-VL-7B | 48.0 | 75.6 | 42.0 | 44.5 | 60.6 | 69.1 | 38.4 | 44.8 | 38.3 | 23.5 | 48.8 | 43.8 | 47.7 | 61.1 | 51.9 | 30.5 | 47.2 |
| Claude3V-Haiku | 47.4 | 74.3 | 44.8 | 51.1 | 63.6 | 67.6 | 26.9 | 46.2 | 35.5 | 22.8 | 50.0 | 35.2 | 42.9 | 54.4 | 69.8 | 34.6 | 38.2 |
| ShareGPT4V-7B | 47.8 | 74.2 | 36.0 | 50.9 | 62.4 | 71.6 | 35.4 | 46.2 | 39.2 | 21.8 | 59.8 | 44.3 | 54.5 | 47.8 | 47.9 | 27.8 | 45.2 |
| LLaVA-v1.5-7B | 46.1 | 72.8 | 34.3 | 47.5 | 61.6 | 68.1 | 34.0 | 46.6 | 36.0 | 22.2 | 58.0 | 42.5 | 57.6 | 45.0 | 40.8 | 26.1 | 44.8 |
| Qwen-2.5-VL-3B | 56.3 | 78.7 | 40.3 | 42.8 | 72.5 | 83.6 | 46.2 | 53.0 | 40.8 | 32.5 | 71.3 | 47.5 | 48.4 | 75.0 | 70.0 | 56.8 | 42.5 |
| Ours (Qwen-3B) | 56.5 | 80.1 | 47.3 | 44.7 | 72.8 | 84.1 | 47.1 | 53.5 | 46.6 | 32.7 | 71.3 | 48.3 | 57.6 | 68.8 | 58.4 | 50.6 | 40.0 |
| *vs. Baseline* | **+0.2** | **+1.4** | **+7.0** | **+1.9** | **0.3** | **+0.5** | **+0.9** | **+0.5** | **+5.8** | **+0.2** | **+0.0** | **+0.8** | **+9.2** | **-6.2** | **-11.6** | **-6.2** | **-2.5** |

## 4.3 MULTIMODAL UNDERSTANDING EVALUATION

**Caption quality evaluation by commercial LLMs.** As shown in Tab. 4, our understanding model (using Qwen-2.5-VL-3B as the baseline) attains high average win rates: **94.7** vs. Show-o, **71.4** vs. OmniGen2, **64.3** vs. Bagel, and **76.3/71.5** vs. Qwen-2.5-VL (3B/7B), while remaining competitive with GPT-4o (47.2). The cross-judge agreement suggests our captions improve along multiple axes, completeness, attribute binding, relational and spatial fidelity, precisely the properties rewarded by the reconstruction-driven training signal.

**Improving the understanding model as a better captioner suitable for generation.** Under the Unified-Bench "caption→generate→compare" protocol, captions produced by our trained understanding model yield the highest reconstruction similarity across all four backbones (Tab. 5): **90.50** (CLIP), **94.35** (LongCLIP), **81.98** (DINO-v2), **77.54** (DINO-v3), with **86.09** Overall. These results indicate that the caption generated by our understanding model is more suitable for generation.

**Evaluation on the multimodal understanding benchmark.** We evaluate on *MMT-Bench* (Ying et al., 2024), which comprises high-level meta-tasks—VR (Visual Recognition), Loc (Spatial Localization), OCR (Text Reading), Count (Object Counting), HLN (Hallucination), IR (Image Retrieval), 3D, VC (Visual Caption), VG (Visual Grounding), DU (Document Understanding), AR (Action Recognition), PLP (Pixel-Level Perception), I2IT (Image-to-Image Translation), RR (Relation Reasoning), Emo (Emotion), and VI (Visual Illusion). The overall score remains essentially unchanged with a marginal improvement over the baseline (+0.2%; Tab. 6). However, if we zoom in to observe *fine-grained visual recognition* suite (Tab. 7), the benefits of our generation-augmented training for perception become pronounced: we observe large absolute gains in Small Object Detection (+40.0%) and Person Re-ID (+60.0%), yielding a +24.4% increase in the fine-grained overall. These results indicate that **generation does not inherently harm understanding, but can instead notably enhance fine-grained visual perception capability.**

## 4.4 ABLATION ON PRE-TRAINING AND POST-TRAINING

We disentangle the effects of two stages: (i) pre-training on our proposed 700k long-context dataset and (ii) post-training with the *unified-GRPO* algorithm. Pre-training primarily benefits the *generation* side, lifting GenEval from 0.71 (Baseline-Decoder) to 0.82 and GenEval++ from 0.296 to 0.401 (reported gains of +11% and +10.5 respectively), while also establishing a strong cross-modal linkage with a Unified-Score of 0.808. Building on this, *unified-GRPO* yields consistent, broad improvements across both *understanding* and *generation*: visual perception metrics rise markedly relative to the Baseline-Encoder (e.g., Small-Obj: 0.45, +40%; ReID: 0.75, +60%), and generation further improves to 0.86/0.475 on GenEval/GenEval++ (+4%/+7.4% vs. Baseline-Decoder). Crucially,

Table 7: **Evaluation results on fine-grained visual perception oriented sub-tasks** on MMT-Bench (Ying et al., 2024). Accuracy is the metric, and the Overall score is computed as the mean of all displayed subtasks. We show notable improvements across various fine-grained understanding tasks, highlighting the positive impact of generation on understanding.

| Model | Overall | Fine-grained Visual Recognition | | | | | Color and Geometry Perception | | | | |
|---|---|---|---|---|---|---|---|---|---|---|---|
| | | Salient Obj. Detection RGBD | Transparent Object Det. | Small Object Detection | Rotated Object Detection | Person Re-ID | Color Constancy | Color Assimilation | Geometrical Relativity | Geometrical Perspective | Polygon Localization |
| InternVL-Chat-V1.2-34B | 63.4 | 28.5 | 66.5 | 64.5 | 46.7 | 60.0 | 34.5 | 44.5 | 82.5 | 75.0 | 46.1 |
| Qwen-VL-Plus | 62.3 | 44.5 | 47.5 | 59.5 | 60.0 | 50.5 | 47.5 | 29.0 | 58.3 | 43.0 | 63.8 |
| GPT-4V | 62.0 | 42.0 | 56.5 | 52.0 | 79.0 | 49.0 | 65.0 | 24.7 | 43.3 | 35.7 | 66.0 |
| GeminiProVision | 61.6 | 45.0 | 38.5 | 43.0 | 50.0 | 72.5 | 38.9 | 53.5 | 46.0 | 43.3 | 36.0 |
| DeepSeek-VL-7B | 53.2 | 40.0 | 53.5 | 43.5 | 36.7 | 32.5 | 27.5 | 52.0 | 54.2 | 56.0 | 23.4 |
| Claude3V-Haiku | 52.2 | 43.0 | 19.5 | 44.0 | 46.7 | 35.0 | 38.5 | 58.5 | 55.8 | 56.5 | 66.7 |
| ShareGPT4V-7B | 51.5 | 40.5 | 39.0 | 37.5 | 27.8 | 24.0 | 52.8 | 26.5 | 60.0 | 65.8 | 32.0 |
| LLaVA-v1.5-7B | 49.5 | 37.5 | 40.0 | 31.5 | 30.0 | 23.0 | 56.9 | 28.0 | 64.0 | 70.0 | 34.0 |
| Frequency | 31.7 | 26.0 | 26.0 | 27.5 | 28.9 | 30.0 | 52.8 | 51.0 | 50.5 | 53.3 | 31.5 |
| Random | 28.5 | 28.5 | 29.0 | 27.0 | 24.4 | 26.0 | 48.6 | 50.0 | 50.5 | 51.7 | 27.5 |
| Qwen-2.5-VL-3B | 32.5 | 25.0 | 15.0 | 5.0 | 33.3 | 15.0 | 28.6 | 50.0 | 60.0 | 58.3 | 35.0 |
| Ours (Qwen-3B) | 56.9 | 45.0 | 45.0 | 45.0 | 55.6 | 75.0 | 42.9 | 60.0 | 65.0 | 75.0 | 60.0 |
| *vs. Baseline* | **+24.4** | **+20** | **+30** | **+40** | **+22.3** | **+60** | **+14.3** | **+10** | **+5** | **+16.7** | **+25** |

Table 8: **Ablation study on the proposed pre-training and post-training strategies.** † refers to the methods using the LLM rewriter. "×" indicates the model is incapable of performing the task, and "−" denotes that the model is frozen during training, thus unchanged performance.

| Model | Visual Perception | | | | Visual Generation | | Unification |
|---|---|---|---|---|---|---|---|
| | Small-Object-Det. | Color-Con. | ReID | Polygon | GenEval | GenEval++ | Unified-Score |
| *Baseline* | | | | | | | |
| Baseline-Encoder | 0.05 | 0.28 | 0.15 | 0.35 | × | × | × |
| Baseline-Decoder | × | × | × | × | 0.71 | 0.343 | × |
| *w/ pre-training* | | | | | | | |
| Ours (*w/* stage-1) | 0.05 | 0.28 | 0.15 | 0.35 | 0.82 | 0.401 | 0.808 |
| *vs. Baseline* | − | − | − | − | **+11%** | **+5.8%** | − |
| *w/ Post-training* | | | | | | | |
| Ours (*w/* stage-1,2,3) | 0.45 | 0.42 | 0.75 | 0.60 | 0.86 | 0.475 | 0.861 |
| *vs. Stage-1* | **+40%** | **+14.3%** | **+60%** | **+25%** | **+4%** | **+7.4%** | **+5.3%** |

*Unified-GRPO* strengthens the mutual promotion between understanding and generation, reflected by a higher Unified-Score (0.861, +5.3% over the pre-trained model), indicating that our proposed reconstruction-based RL objective can serve as a unified goal to optimize both.

## 5    LIMITATION AND FUTURE WORK

In our experiment, we observe an unexpected decrease in understanding performance on *text-related recognition tasks* (Tab. 6), with accuracy dropping by approximately 10% on Document Understanding (DU) and OCR. We attribute this to the inherent limitations of current generation models, which *struggle with accurate text rendering*, a well-known challenge in generative modeling (Ramesh et al., 2022). We hypothesize that our generation model may provide "misleading" reward signals during training (fail to reconstruct the correct text), which negatively affects the encoder's recognition of textural content. This suggests that, conceptually, understanding and generation should ideally benefit one another, but the *gains are now mainly constrained by the "imperfections" of the generation component.* We acknowledge this limitation and leave its mitigation for future work.

## 6    CONCLUSION

We show that an Auto-Encoder is a viable core for unifying multimodal understanding and generation. Building on this idea, we introduce **UAE**, which warms up the decoder on long-context captions. We then propose Unified-GRPO, a new RL-based post-training method that jointly optimizes caption informativeness and reconstruction fidelity. To quantify progress toward unification, we present Unified-Bench, the evaluation tailored to the bidirectional nature of UMMs. During training, we observe an "aha moment": captions become longer and more precise while reconstructions sharpen, evidencing coherent, bidirectional information flow. Extensive experiments demonstrate that the generation and understanding can truly benefit together to a certain extent but are still constrained by the scope of the generation model. Together, these efforts offer a clear recipe and measurement protocol for building truly unified multimodal models.

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

APPENDIX OVERVIEW

## A  RELATED WORK

**Unified Multimodal Generation and Understanding.**   Recent advancements in multimodal AI have led to the development of Unified Multimodal Models (UMMs), a new class of frameworks that integrate both perception and generation capabilities across modalities through a unified, end-to-end trainable architecture (Zhang et al., 2025). The architectural designs of current UMMs can be broadly categorized into two paradigms: **(1) AR-based Approaches**: In this setup, all modalities, including images and text, are tokenized and processed sequentially using an autoregressive transformer. Systems like Chameleon and EMU generate image tokens akin to language modeling by predicting the next token in a sequence (Team, 2024; Qu et al., 2024; Wu et al., 2025a; Chen et al., 2025b; Wu et al., 2024; Li et al., 2024c). An evolution of this idea is seen in Show-o (Xie et al., 2024a), which enhances token prediction with a discrete diffusion mechanism, introducing a structured denoising process during generation. **(2) Hybrid AR-Diffusion Architectures:** Some models combine autoregressive modeling with diffusion-based image synthesis Yan et al. (2025). For instance, Transfusion and similar systems (Zhou et al., 2024; Deng et al., 2025; Ma et al., 2024; Shi et al., 2024; Xie et al., 2025) extend a shared transformer backbone with a dedicated diffusion or flow-matching head for high-fidelity image generation. Alternatively, other approaches freeze a pre-trained MLLM and use learnable query modules or MLPs to extract and route intermediate representations to an external image generator (Pan et al., 2025; Chen et al., 2025a; Lin et al., 2025). A more recent direction integrates standard autoregressive language processing with masked-autoregressive reconstruction for visual data. MAR (Li et al., 2024d) enables image generation without relying on vector quantization, instead reconstructing patches in a flexible order. This approach has been adopted in models such as Harmon (Wu et al., 2025c; Fan et al., 2025; Wang et al., 2025a). Meanwhile, some works (Geng et al., 2025; Chen et al., 2025a) use a discretized SigLIP (Tschannen et al., 2025) to convert images into tokens, training a single autoregressive model over these visual and language tokens, while employing a diffusion model for the final image decoding.

**Reinforcement Learning in Generative Models.**   The widespread success of Reinforcement Learning from Human Feedback (RLHF) in aligning large language models (LLMs) with human intent (Christiano et al., 2017; Hu et al., 2022) has inspired its application to text-to-image generation. In this context, a common strategy involves first training a reward model (RM) that learns from human judgments—either general aesthetic preferences (Xu et al., 2024) or alignment between prompts and generated images (Xu et al., 2023), followed by reinforcement learning to optimize the generative model accordingly (Black et al., 2023). Despite its promise, this two-stage approach faces significant limitations when applied to image editing tasks. Reward models are often brittle and challenging to design robustly (Miao et al., 2024), and they can be gamed through superficial changes that maximize reward without improving actual quality—a phenomenon known as "reward hacking" (Wang et al., 2025b). More recently, alternative optimization frameworks like GRPO (Shao et al., 2024) have emerged as viable solutions, demonstrating effectiveness in tuning both diffusion and flow-matching based models. Extensions such as FlowGRPO (Liu et al., 2025) and DanceGRPO (Xue et al., 2025) illustrate the adaptability of these algorithms to complex generative processes, offering a more stable and fine-grained path toward aligning visual outputs with human expectations—particularly in dynamic, iterative editing scenarios where traditional methods fall short.

**Benchmarking Multimodal Understanding, Generation, and Unification.**   Evaluating unified multimodal models (UMMs) typically involves aggregating performance across multiple specialized

benchmarks, each targeting distinct capabilities. For assessing visual understanding, widely adopted benchmarks include ScienceQA (Lu et al., 2022), MMMU (Yue et al., 2024), VQA (Antol et al., 2015), GQA (Hudson & Manning, 2019), and MM-Bench (Liu et al., 2024), all of which rely heavily on large-scale datasets with human-annotated images and labels. In contrast, our proposed UniBench introduces a novel paradigm as a VQA-style benchmark specifically designed for *generated* images, eliminating the dependency on real-image annotations by evaluating comprehension directly on synthesized content. For generative capability assessment, image quality is commonly measured using metrics such as FID (Heusel et al., 2017), ImageReward (Xu et al., 2023), and LIQ (Tian et al., 2025), often evaluated on standard image corpora like MSCOCO (Lin et al., 2014) or LAION-5B (Schuhmann et al., 2022). Additional factors such as text-image alignment (Hessel et al., 2021), fairness (Lee et al., 2023), and stylistic consistency (Peng et al., 2024) are also considered, drawing from benchmarks like HRS (Bakr et al., 2023). However, unified models place greater emphasis on instruction-following and coherent joint reasoning across perception and generation. As such, evaluation frameworks tailored to text-to-image synthesis, such as GenEval (Ghosh et al., 2023), DPG-Bench (Hu et al., 2024), and T2I-CompBench++ (Huang et al., 2025), which are particularly relevant. These assess fine-grained attributes including object presence, spatial relations, counting accuracy, color fidelity, and positional reasoning (Bakr et al., 2023; Li et al., 2024a; Cho et al., 2024). Despite their utility, existing benchmarks are not specifically designed for the dual perception-generation nature of UMMs, leaving a gap in comprehensive, integrated evaluation. To address world-knowledge grounding in image synthesis, WISE (Niu et al., 2025) was recently introduced to evaluate models' implicit understanding of real-world constraints across domains such as food preparation, material physics, and object affordances. More recently, UniEval (Li et al., 2025) proposes a new benchmark dedicated to unified multimodal modeling, covering a broader range of semantic, structural, and logical challenges with increased task difficulty and potential for model improvement.

## B DATASET DETAILS

**SFT (long-context T2I).** We construct a large-scale, high-quality text–image pre-training corpus consisting of approximately 700k image-caption pairs to effectively warm up the diffusion transformer decoder for long-context understanding and generation. Each sample in this dataset pairs a exceptionally detailed natural language caption with an image paired with a 1024×1024 resolution image with an exceeding 250 English words in length. These captions are generated using InternVL-78B (Chen et al., 2024), a state-of-the-art vision-language model, over a diverse private collection of images curated to cover a broad range of scenes, including urban landscapes, indoor environments, human activities, object interactions, and complex multi-subject compositions. The captioning process is specifically optimized to emphasize fine-grained descriptions of objects, their visual attributes (e.g., color, texture, material, shape), spatial relationships (e.g., "a red backpack rests on the wooden bench beneath a streetlamp"), and global scene layout (e.g., lighting conditions, depth cues, foreground-background structure). This ensures that the textual input contains the rich semantic structure necessary for training the model to map extended linguistic descriptions into coherent visual outputs. Two representative examples from this dataset are illustrated in Fig. 8 and Fig. 9, showcasing both the complexity of the imagery and the descriptive density of the corresponding captions. During the supervised pretraining phase, the full long-form caption is used as the input prompt, and the diffusion transformer is trained end-to-end to denoise and generate the matching high-resolution image, thereby learning precise alignment between nuanced textual semantics and pixel-level visual details.

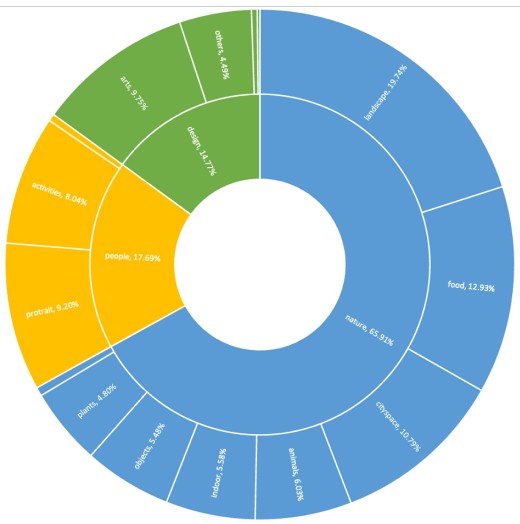

Figure 5: The illustration of the distribution of our proposed 700k long-context dataset.

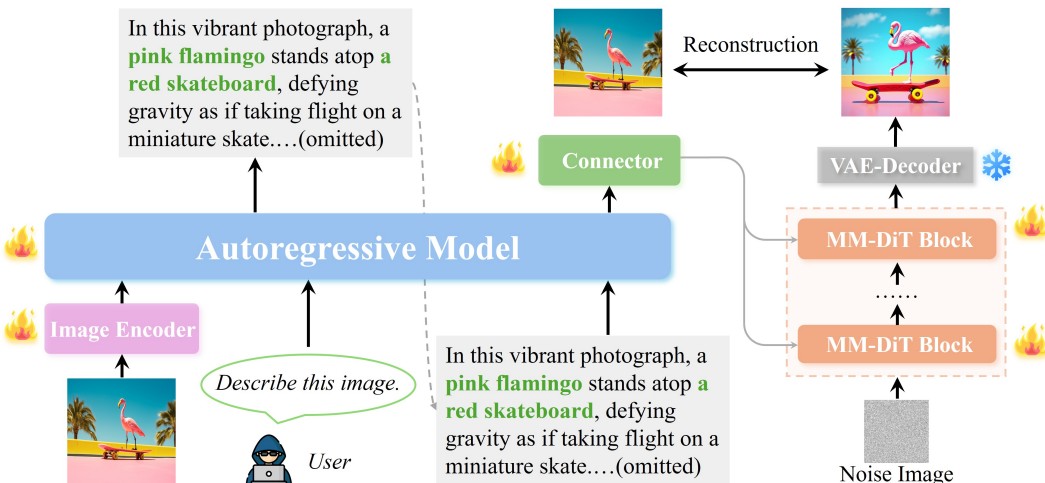

Figure 6: The detailed illustration of our framework design. Our framework employs an autoregressive LVLM to process the input image embedding derived from the original image. The model generates a text caption, which is then fed into the autoregressive LLM. From this, we extract the final hidden state and project it through a connector into the decoder's feature space, where it serves as the condition for image generation.

**GPT-4o distillation (50K).** To further enhance the linguistic quality, coherence, and stylistic consistency of the training captions, particularly for challenging or ambiguous visual content, we perform knowledge distillation using GPT-4o (OpenAI, 2025). This 50K data is a part of our total 700k data. Leveraging the auto-script pipeline introduced in GPT-ImgEval (Yan et al., 2025), we select a subset of 50,000 particularly complex or semantically dense images from our broader corpus and re-generate their captions through carefully designed prompting strategies that encourage narrative fluency, logical flow, and comprehensive coverage of visual elements. These distilled captions average around 300 words and exhibit superior grammatical correctness, richer vocabulary, and more consistent syntactic structure compared to the base InternVL-generated annotations. Importantly, they also demonstrate improved reasoning about occluded objects, inferred actions, and contextual implications (e.g., weather, time of day, emotional tone). These 50K high-fidelity text–image pairs are then integrated into the main pretraining mix with elevated sampling weight, serving as semantic anchors that guide the model toward generating images with greater anatomical accuracy, environmental plausibility, and adherence to subtle instruction cues embedded in long prompts.

**RL stage data (1K).** For the reinforcement learning (RL) phase, we curate a compact but highly refined dataset of 1,000 real-world photography images selected for exceptional compositional quality, visual clarity, and semantic richness. These images span diverse domains such as portrait photography, architectural shots, nature scenes, and dynamic street photography, all captured under realistic lighting and perspective conditions. In addition to these hand-picked photographs, we incorporate a specialized subset of synthetic yet photorealistic data from Echo-4o (Ye et al., 2025), which provides tightly aligned text-image pairs with expert-level captions and controlled visual variations. This combined RL dataset is used in a reconstruction-driven optimization framework: given a caption derived from one of these target images, the model is tasked with generating a new image, and its output is evaluated against the original using a learned reward model that assesses fidelity, detail preservation, and semantic alignment. Through this closed-loop paradigm, improved captioning leads to better reconstruction, which in turn refines generation capabilities.

**Data for evaluation in Unified-Bench.** To evaluate the model's performance on the proposed Unified-Bench, we randomly sample 100 images from the LAION-5B dataset (Schuhmann et al., 2022) to serve as a dedicated test split. These images are selected without any filtering or curation based on content or aesthetic score, ensuring a representative and unbiased distribution across categories, styles, and complexity levels.

## C  TRAINING SETTINGS

**Custom settings for Stable-Diffusion**  In this work, we employ Stable-Diffusion-3.5-large (Esser et al., 2024) as the image decoder to generate high-quality images from textual inputs. To enhance long-context understanding and inject richer world knowledge into the generation process, we replace the original T5 text encoder of Stable Diffusion with the powerful vision-language text encoder from Qwen-2.5-VL, following a strategy similar to UniWorld-V1 (Lin et al., 2025); consequently, the T5 encoder is no longer used in our pipeline. Additionally, we retain the original CLIP text encoder of Stable Diffusion solely for producing unconditional (null) embeddings: during both training and inference, an empty prompt (i.e., a blank or null string) is passed through this CLIP encoder to obtain the corresponding text embedding for classifier-free guidance (CFG). During inference, we set the CFG scale to 5.0 and use 40 denoising steps for the sampling process, same to the settings used in FlowGRPO (Liu et al., 2025). **LoRA-Adaptation.** Following previous works (Liu et al., 2025; Dettmers et al., 2023; Yan et al., 2024; Jin et al., 2024), we apply LoRA (Hu et al., 2022) adaptation for both the encoder and decoder for RL post-training, as it can help preserve the rich semantic knowledge learned from pre-training while efficiently and effectively learning novel knowledge from the new task. We maintain the same settings of LoRA with Flow-GRPO (Liu et al., 2025).

**Training details of stage-1.**  For long-context pretraining, we conduct large-scale training using 8 H800 nodes, each equipped with 8 GPUs, resulting in a total of 64 GPUs. The full pretraining phase is performed on these 8 nodes, while the subsequent reinforcement learning (RL) stage utilizes a reduced setup of 4 nodes (32 GPUs). During the initial pretraining phase, the model is trained for 10,000 steps at a resolution of $512 \times 512$ with a batch size of 32 and a learning rate of $2 \times 10^{-4}$. This is followed by an additional fine-tuning phase of 5,000 steps at a higher resolution of $1024 \times 1024$, using a smaller batch size of 16 and a reduced learning rate of $8 \times 10^{-5}$, while continuing to train on the same dataset to refine image fidelity and detail generation. We employ the AdamW optimizer Loshchilov & Hutter (2017) with standard hyperparameters: $\beta_1 = 0.9$, $\beta_2 = 0.999$, and $\epsilon = 1 \times 10^{-8}$. To enable efficient distributed training with large models and batch sizes, we implement the ZeRO-2 parallelization strategy (Rajbhandari et al., 2020), which significantly reduces memory consumption across devices while maintaining high computational throughput.

**Training details of stage-2.**  In Stage-2, we employ the GRPO RL algorithm (Shao et al., 2024) to fine-tune the LLM while keeping the corresponding visual encoder frozen. We empirically observe that updating the visual encoder during RL training can lead to instability and degradation in image quality such as anomaly artifacts, structural collapse, or semantic inconsistency, so we disable its gradient updates to preserve visual feature integrity. To enable effective sampling for RL-based image generation, we treat the combination of the DiT and a pre-trained CLIP model (Radford et al., 2021) as a unified, frozen reward module. This composite model operates purely in inference mode: given a generated image and its corresponding reconstructed image from the input caption, it computes a similarity score that serves as the final reward signal in the GRPO framework. Specifically, we adopt LongCLIP (Zhang et al., 2024) in place of the standard CLIP encoder, as our setting involves significantly longer textual contexts, often exceeding hundreds of words, whereas standard CLIP is limited to 77 tokens, making it insufficient for capturing extended descriptions. LongCLIP's extended context capacity allows more accurate alignment between long-form captions and generated images, thereby providing more reliable and nuanced rewards. During training, we use a learning rate of $1 \times 10^{-6}$ and a batch size of 1 due to the high computational cost of diffusion-based RL. For each prompt, we generate 4 sampled images to estimate the policy gradient in GRPO, and we set the KL regularization coefficient $1 \times 10^{-6}$, indicating that we do not apply any penalty for divergence from the reference policy (i.e., no KL control), focusing solely on reward maximization. The temperature of LLM is set to be 1.0. The prompt used to do the LLM sampling is shown below.

---

**Prompt for the LLM Sampling**

**System Prompt:** You are an expert vision-language model.
**User Prompt:** Your task is: Given an input image, generate a **textual description** of the image. If there is text in the image, transcribe it inside double quotes.

Now, carefully analyze the input image and output the full description.

Input Image: {{image_path}}

---

Note that we do **not** explicitly require the LLM to generate descriptive or comprehensive captions during training. After RL, the LLM autonomously produces longer and richer captions that are more conducive to high-fidelity image generation, even though no explicit supervision or loss is applied to the caption content itself. This emergent behavior suggests that the ***RL signal from image reconstruction quality implicitly guides the LLM toward generating more detailed and image-friendly textual descriptions.***

**Training details of stage-3.** Similar to Stage-2, in Stage-3, we also employ the GRPO objective for optimization, but with a key difference: we use it to finetune the DiT while keeping the entire LVLM frozen, including both the LLM and the visual encoder. To introduce stochasticity during sampling, inspired by FlowGRPO Liu et al. (2025), we adopt an SDE-based (Stochastic Differential Equation) noise sampling strategy, which injects diverse random noise trajectories into the diffusion process and enhances exploration during training. Specifically, we fix the caption output from the frozen LLM for each input prompt, and for every such caption, the DiT generates 8 different images to estimate the policy advantage in GRPO. We set the global batch size to 4 (accumulated over multiple steps if necessary), and use a KL regularization coefficient $\beta = 0.01$ to mildly constrain the DiT's output distribution from deviating too far from the reference generator, thus improving training stability. For computational efficiency, we use 20 denoising steps during training, while increasing to 30 steps during validation to obtain higher-quality samples for evaluation. The prompt used to generate caption from LLM is the same with the Stage-2.

## D    QUALITATIVE EXAMPLES

**GenEval++ visualizations.** Fig. 11 presents six representative prompts from **GenEval++** (Ye et al., 2025), where each instruction contains *three or more* entities with distinct attributes and spatial relations. Across these examples, UAE shows three recurring strengths. First, it preserves *attribute binding* under multi-entity scenes: for "three *purple* hair dryers and one *pink* surfboard," UAE attaches colors to the correct categories without leakage, whereas baselines often color a surfboard purple or mix pink/purple across objects. Second, UAE is more reliable on *discrete counts* while respecting co-occurring constraints: for "three beds on the above and three parking meters on the below," UAE maintains the 3+3 cardinality *and* the vertical arrangement; competing models tend to be off-by-one or satisfy the layout but drop a meter/bed. Third, UAE handles *left/right and grouping* more faithfully: for "an *orange* laptop on the *left* and a *purple* knife on the *right*," our outputs keep the polarity and avoid color–object swaps that are common failure modes. Similar advantages emerge in the "two cows, two books, and one donut" and "six vases" prompts: UAE balances global composition with local details, maintaining counts while rendering plausible object geometry and material. These observations align with Tab. 3: UAE leads on *Color/Count* and *Pos/Count*, and is competitive on *Color/Pos* and *Multi-Count*, reflecting robust satisfaction of *joint* constraints rather than excelling on a single dimension.

**Enhancing model's comprehensive perception by the generation model.** Fig. 12 contrasts captions used for reconstruction on a challenging example (small black dog wearing a yellow beanie and glasses). Baselines reveal three typical errors. (i) *Category drift*: some misidentify the subject as a monkey, causing the generator to synthesize an incorrect species. (ii) *Attribute omissions or swaps*: descriptions drop key items (beanie, glasses) or mismatch apparel colors, leading to reconstructions that caricature the outfit. (iii) *Under-specified scenes*: vague backgrounds and missing lighting cues prevent consistent photographic style at inference. UAE's caption, in contrast, enumerates the full set of semantics—species, apparel *type and color*, eyewear, pose, occlusions ("ears are not visible"), background style ("blurred, park-like"), and lighting—producing a reconstruction that preserves identity, attire, and overall aesthetic. This example typifies the mechanism by which better understanding (denser, better-bound descriptions) yields better generation, echoing our Unified-Bench gains in Tab. 5.

**Prompt for Generating Long-Context Image Captions**

**System Prompt:**
You are a top-tier visual descriptive author. Your mission is to generate a comprehensive, high-information-density, and meticulously detailed text description for a given image. Your description must achieve sufficient precision and detail to support a state-of-the-art text-to-image model in accurately and faithfully reconstructing the original image. You must strictly adhere to all of the following format requirements and detailed guidelines.

**User Prompt:**
*Task Description:*
Please carefully observe and analyze the following image, and generate a high-fidelity description in accordance with the established rules and format requirements.

*Description Guidelines:*
1. Subject Identification: Clearly identify all core subjects (e.g., persons, vehicles, text). If a public figure is involved, they must be accurately named. The race, estimated age, and facial expression should be described if visually discernible and can be determined with high confidence.
2. Exhaustive Detail Features: Provide a complete inventory of visible attributes for every subject and element. This includes, but is not limited to: precise colors, materials, textures, and shapes (e.g., the specific way bangs fall, the flow of hair at the back). If text is present, it must be transcribed verbatim, and its typography (font, style, color, size) and exact location must be described.
3. Quantity & Spatial Positioning (Critical Requirement): This is of utmost importance. The quantity of all subjects and their spatial arrangement must be precisely stated. A combination of absolute positioning (e.g., "in the left side of the frame") and relative positioning should be used, especially when multiple subjects have a relational layout (e.g., "Subject A is to the left of Subject B, who is to the left of Subject C").
4. Kinetics, Posture & Environment: Describe the subject's behavior or actions with extreme granularity (detailing the posture of each arm and leg). In parallel, provide a detailed description of the background scene and the overall atmosphere of the image (e.g., "a city skyline under backlighting").
5. Artistic Style & Technique: Identify the specific style (e.g., anime, photorealistic, Makoto Shinkai-style), lighting effects (e.g., soft light, Tyndall effect), and shot type, including the overall camera perspective (e.g., top-down view, low-angle shot, centered composition, symmetrical composition).
6. Language Requirements: Maintain objectivity throughout the description. Describe only visible content. Avoid speculation (e.g., "they might be a couple") or introducing irrelevant external information (e.g., historical context).
7. Conciseness & Informational Density: Use precise and concise language to cover all core information, avoiding redundancy. The goal is to maximize the information-to-word ratio while ensuring grammatical correctness and a logical structure.

*The output format:*
1. Overall Structure: The entire description must be consolidated into a single, cohesive, and complete paragraph.
2. Opening Summary: The first sentence of the paragraph must be a high-level summary that encapsulates the entire scene, setting, and core subject(s).
3. Hierarchical Description: If the image can be clearly divided into multiple logical regions (e.g., foreground, midground, background, or multiple scenes), each region should be described sequentially and independently within the single-paragraph structure.
4. Word Count: The total description must be no less than 500 words.

*Reference Examples:*
Example 1: {{example-1}}
Example 2: {{example-2}}

Figure 7: Illustration of the reconstruction results when unfreezing ViT (the visual encoder of the MLLM) for joint training. We observe that the generated output collapses, semantically important details such as "two pumpkins" and "one candle" are missing. This degradation motivates us to keep the ViT frozen during finetuning across all experiments.

---

**Prompt Used to Perform LLM Judge for Caption Quality**

**User Prompt:**
You will conduct a multi-dimensional analysis of each caption based on the specific criteria listed below. For each criterion, you will assign a score from 1 (very poor) to 10 (excellent). After scoring, you must provide a detailed, structured comparative analysis and declare a final winner.

**Evaluation Criteria & Scoring:**
Please evaluate each caption against the following four criteria. Provide your scores in a markdown table.

1. **Comprehensiveness, Descriptive Richness, and Accuracy:**
   - How deeply does the caption describe the image? Does it go beyond a superficial glance to include important, specific details (e.g., colors, textures, materials, lighting, background elements, expressions)?
   - Does it effectively and accurately describe the context (e.g., a black dog not a monkey, or brown eyes not black), environment (background description)?
   - Does the caption capture subtle nuances that a casual observer might miss?

2. **Linguistic Fluency and Naturalness:**
   - Is the caption grammatically correct and well-written in natural-sounding English?
   - Does it flow like a human would describe the scene, or does it sound robotic, disjointed, or like a list of keywords?
   - Is the vocabulary choice sophisticated, appropriate, and engaging?

3. **Semantic and Compositional Insight:**
   - Does it effectively capture and convey the overall mood, atmosphere, emotion, or narrative implied by the scene?
   - Does it demonstrate an understanding of the image's composition (e.g., what is in the foreground vs. background)?

Based on the above rules, provide a comprehensive, head-to-head comparison of the two captions. Structure your analysis with subheadings for each of the four criteria. For each criterion, explicitly quote phrases from both Caption A and Caption B to illustrate your points and justify the difference in their scores. Explain not just *what* is different, but *why* one caption's approach is superior for describing the provided image.

Finally, please declare the winner based on your detailed comparative analysis above. This section must contain only a single letter.

**Final Answer:** [A or B].

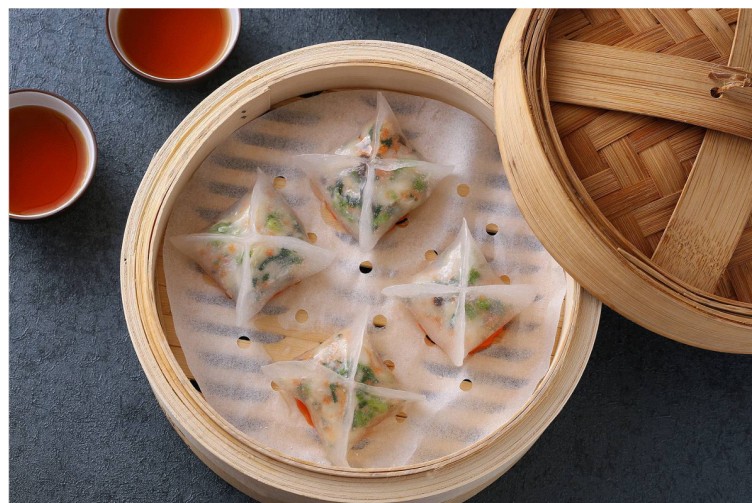

This is an image showcasing traditional Chinese dim sum, with a warm color palette dominated by bamboo steamers and a dark background, creating a rustic and homely atmosphere.

### Overall Composition and Angle
- **Photography Angle**: Top-down with a slight tilt, allowing clear visibility of the details and placement of each dim sum.
- **Background Material**: A dark, rough tabletop paired with a light bamboo steamer, giving a warm and rustic tone.
- **Composition**: The steamer is at the center with four dim sum pieces. Two small bowls of tea are placed in the upper left corners, creating a symmetrical and balanced visual.

### Main Food Components
1.  **Dim Sum (Center)**
 - **Quantity and Position**: Four translucent dumplings in a diamond shape, evenly placed in the steamer with appropriate spacing.
- **Shape and Color**: The dumplings have a transparent outer skin, diamond-shaped, with colorful fillings including green vegetables, orange carrots, and white meat.
- **Material and Details:** The skin is thin and transparent, clearly showing the filling inside, with a smooth and slightly glossy surface.
2. **Tea Bowls (Upper Left Corners)**
- **Quantity and Position:** Two small tea bowls located in the upper left and lower left corners.
- **Shape and Color:** The bowls are round with a dark brown exterior and white interior, filled with clear, orange-red tea.
- **Material and Details**: The bowls are ceramic with a smooth surface, and the tea has a rich color, indicating its aroma and texture.

### Background and Environment
- **Background Material**: A dark, rough tabletop that contrasts sharply with the bamboo steamer, highlighting the dim sum.
- **Steamer Details**: Bamboo steamer with a partially open lid revealing the dim sum inside. The bottom of the steamer is lined with white baking paper, marked with bamboo patterns.

### Artistic Style and Techniques
- **Style**: Realistic, focusing on detail and texture representation.
- **Lighting Effects:** Soft, even lighting on the dim sum and tea bowls, emphasizing texture and color without harsh shadows.
- **Perspective and Angle**: Top-down view, making the composition clear and allowing full visibility of details.

### Language Standards
- **Objective Description**: Only visible content is described, without speculation or unrelated information.

### Conciseness
- **Precise Description**: Core information is covered concisely, avoiding redundancy. To accurately recreate this image, ensure the following elements:
- A central bamboo steamer with four translucent diamond-shaped dumplings with colorful fillings.
- Two small tea bowls with dark brown exteriors and white interiors, containing orange-red tea, placed in the upper left corners.
- A dark, rough tabletop background contrasting with the bamboo steamer to highlight the dim sum.
- Soft, even lighting to emphasize the texture and color of the food.

Figure 8: Visual example of the proposed 700k long-context text-to-image dataset.

# E  ADDITIONAL EXPERIMENTAL RESULTS

**The text-to-image generation results on DPG-Bench.**  On DPG-Bench (Tab. 9), UAE achieves the top scores on *Entity* (91.43), *Attribute* (91.49), and *Relation* (92.07), and ranks second overall with **84.74**, closely trailing Bagel (85.07). The sub-score pattern suggests UAE's advantages come

Table 9: Comparisons of text-to-image generation ability on DPG-Bench (Hu et al., 2024) benchmark. **Bold** indicates the best result, and underlined denotes the second best.

| Method | Global | Entity | Attribute | Relation | Other | Overall |
|---|---|---|---|---|---|---|
| **Dedicated T2I** | | | | | | |
| SDXL Podell et al. (2023) | 83.27 | 82.43 | 80.91 | 86.76 | 80.41 | 74.65 |
| PlayGroundv2.5 Li et al. (2024b) | 83.06 | 82.59 | 81.20 | 84.08 | 83.50 | 75.47 |
| Hunyuan-DiT Li et al. (2024e) | 84.59 | 80.59 | 88.01 | 74.36 | 86.41 | 78.87 |
| PixArt-$\Sigma$ Chen et al. (2023) | 86.89 | 82.89 | 88.94 | 86.59 | 87.68 | 80.54 |
| DALLE3 OpenAI (2024) | **90.97** | 89.61 | 88.39 | 90.58 | 89.83 | 83.50 |
| SD3-medium AI (2024) | 87.90 | 91.01 | 88.83 | 80.70 | 88.68 | 84.08 |
| FLUX.1-dev Labs (2024) | 82.1 | 89.5 | 88.7 | 91.1 | 89.4 | 84.0 |
| OmniGen Xiao et al. (2025) | 87.90 | 88.97 | 88.47 | 87.95 | 83.56 | 81.16 |
| **Unified Model** | | | | | | |
| Show-o Xie et al. (2024a) | 79.33 | 75.44 | 78.02 | 84.45 | 60.80 | 67.27 |
| EMU3 Wang et al. (2024b) | 85.21 | 86.68 | 86.84 | 90.22 | 83.15 | 80.60 |
| TokenFlow-XL Qu et al. (2025) | 78.72 | 79.22 | 81.29 | 85.22 | 71.20 | 73.38 |
| Janus Pro Chen et al. (2025b) | 86.90 | 88.90 | 89.40 | 89.32 | 89.48 | 84.19 |
| BLIP3-o 4B Chen et al. (2025a) | - | - | - | - | - | 79.36 |
| BLIP3-o 8B Chen et al. (2025a) | - | - | - | - | - | 81.60 |
| UniWorld-V1 Lin et al. (2025) | 83.64 | 88.39 | 88.44 | 89.27 | 87.22 | 81.38 |
| OmniGen2 Wu et al. (2025b) | 88.81 | 88.83 | 90.18 | 89.37 | **90.27** | 83.57 |
| BAGEL Deng et al. (2025) | 88.94 | 90.37 | 91.29 | 90.82 | 88.67 | **85.07** |
| **UAE** | 83.11 | **91.43** | **91.49** | **92.07** | 84.32 | 84.74 |

from faithful entity grounding and relation handling under long prompts, translating into competitive end-to-end generation quality within a unified architecture.

**Prompt list used in Fig. 10.** We provide the full caption for each sample in generation order, reading from left to right and top to bottom, row by row.

- **Sample-1**. A close-up portrait of a ginger tabby cat, its fur a rich tapestry of warm amber and deep russet stripes that catch the soft, directional light illuminating its face from the side, highlighting the velvety texture of its coat and the subtle contours of its cheekbones, while its large, luminous green eyes gaze intently off-camera with an expression of quiet contemplation and alert curiosity, framed by long, delicate white whiskers and perked ears that suggest attentiveness, all set against a dark, shadowy background that isolates the feline subject and enhances the dramatic, almost painterly quality of the image, emphasizing the cat's regal poise and enigmatic presence.
- **Sample-2**. The building on the left is a light beige color with a series of rectangular windows framed in red, some with small white panes. These windows have simple brick or mortar surrounds and are uniformly spaced, creating a rhythmic pattern across the facade. The ground floor features a small shop area with a white canopy providing shade for outdoor seating. The canopy is supported by metal poles and holds a few tables under its shelter. Behind this canopy, various items can be seen, including a few chairs and tables, indicating a café or small eatery. A white umbrella stands next to the shop entrance, adding to the cozy atmosphere.Above the shop, the building has a series of small balconies with metal railings, each adorned with potted plants and hanging baskets, contributing to the pedestrian-friendly urban design. The ground floor has a mix of business signs, some of which are partially visible but not legible, suggesting a bustling commercial area. There's a dark green signboard affixed to one of the windows, possibly indicating a specialty shop or restaurant. The neighboring building on the right is a lighter shade of beige with a pastel green section near the top. Its windows are similarly framed in red, with larger panes and a more varied arrangement compared to the first building. This building features balconies with metal railings and small rectangular windows. The exterior walls show some wear and tear, with subtle moldings and patches of weathering, adding character to the structures. In front of these buildings lies a cobblestone street, which is partially shaded by the shadows cast by the buildings. A large stone fountain occupies the foreground, its base circular and gray, with a worn, dark surface. The pavement around the fountain is paved with irregularly shaped stones, creating a rustic, old-world feel. The sunlight creates dramatic contrasts, with deep shadows and bright highlights accentuating

the textures of the buildings and the cobblestones. The street is quiet, devoid of people, which enhances the serene and timeless atmosphere of the scene.

- **Sample-3**. A photo of hearty Chinese meal.
- **Sample-4.** This serene watercolor painting evokes the tranquil spirit of a traditional Chinese riverside village, where mist-laden mountains recede into a soft, pale sky, their layered silhouettes rendered in gentle washes of gray and muted green that dissolve into atmospheric haze; along the calm, reflective riverbank, white-walled houses with dark-tiled, upturned eaves nestle among lush trees, their architecture echoing classical Jiangnan aesthetics, while two slender wooden boats glide silently on the glassy water—one closer to the foreground with its simple mast and open cabin, the other a distant speck fading into the fog—imbuing the scene with quiet movement and timeless stillness, as the interplay of light and shadow across the rippling surface and the subtle gradations of ink suggest not only depth and distance but also a meditative harmony between nature and human habitation, capturing the essence of poetic rural life suspended in a dreamlike, almost ethereal moment.
- **Sample-5**. A vibrant blue skateboard with bold, graffiti-style graphics—featuring swirling red and yellow patterns and stylized lettering—stands upright on cracked concrete, its bright red wheels and silver trucks catching the sunlight, casting a sharp shadow on the ground, while in the blurred background, a weathered wall adorned with colorful street art and a partially visible skate ramp hint at an urban skate park setting, blending raw energy with artistic expression under a clear, sunlit sky.
- **Sample-6**. A solitary, gnarled tree with twisted, leafless branches stretches skyward like a skeletal sentinel in the heart of a vast desert landscape, its weathered trunk rooted firmly in the ochre sands that stretch to the horizon, dotted sparsely with low-lying shrubs; above, a dramatic expanse of billowing cumulus clouds drifts across a brilliant blue sky, casting shifting shadows over the arid terrain, while in the distance, the imposing silhouette of red rock mesas rises majestically against the horizon, lending a sense of ancient grandeur and timeless solitude to the scene, where nature's raw resilience and stark beauty are captured in perfect harmony under the vast, open heavens.
- **Sample-7**. A striking traditional East Asian ink painting captures the vibrant essence of a blossoming plum tree, its gnarled, darkly rendered branches—executed with bold, expressive brushstrokes of sumi ink—arching gracefully across the stark white paper to cradle clusters of vivid crimson flowers, each petal delicately shaped with fluid washes of red that convey both vitality and fragility, while subtle hints of green foliage at the lower left suggest the quiet emergence of new life; the composition balances dynamic movement with serene stillness, evoking themes of resilience and renewal as the blossoms defiantly bloom against the void, enhanced by the faint calligraphic inscription near the trunk and the small red seal in the corner, which together anchor the piece in cultural tradition and artistic intention.
- **Sample-8**. In a breathtaking, sun-drenched meadow of lush rolling hills dotted with wildflowers and scattered boulders, a young boy with soft silver-gray hair and wide, awestruck blue eyes gazes upward in wonder as he gently cradles a radiant, living flame between his outstretched palms—a glowing, teardrop-shaped orb of golden-orange fire that pulses with warmth and light, its edges flickering with delicate embers against the backdrop of a brilliant blue sky streaked with fluffy white clouds and distant snow-capped mountains; dressed in a simple light-blue jacket over a crisp white shirt, the child embodies innocence and quiet awe, as if he has just summoned or discovered this mystical force, transforming the idyllic pastoral landscape into a realm where magic feels not only possible but tenderly held, evoking a sense of harmony between nature, wonder, and the boundless imagination of youth.
- **Sample-9**. A vibrant, sun-drenched tropical beach unfolds under a brilliant azure sky dotted with fluffy white clouds, where the crystal-clear turquoise waters gently lap against golden sands lined with swaying palm trees casting dappled shadows on the shore, and at the heart of this serene paradise, the bold, three-dimensional white letters spelling "KEEP CALM" rise majestically from the sea's edge, their clean, modern font contrasting with the organic beauty of nature while reinforcing the tranquil mood, as if the very landscape itself is whispering a soothing mantra of peace, relaxation, and escape from the chaos of everyday life.
- **Sample-10**. A dazzling, multifaceted purple diamond rests regally upon a shimmering bed of iridescent violet sand, its precisely cut facets catching and refracting beams of ethereal light that radiate from behind, casting a luminous glow across the scene and accentuating the gem's deep amethyst hue with flashes of electric violet and cool silver highlights; the background dissolves into a dreamy, softly diffused gradient of lavender and indigo, enhancing the jewel's otherworldly brilliance and making it appear almost suspended in a mystical twilight realm, where every angle

of its polished surface seems to whisper secrets of rare beauty and enchanted allure, evoking both luxury and fantasy in a single, captivating moment.

- **Sample-11**. In a rain-slicked, neon-drenched cyberpunk cityscape at night, a mysterious hooded figure stands silhouetted against a kaleidoscope of glowing skyscrapers and pulsating billboards, their face obscured by shadow as they hold aloft a luminous rectangular sign that boldly proclaims "UAE" in vibrant, electric-blue neon lettering, casting an otherworldly glow on their gloved hands and the wet pavement below, where reflections of magenta, cyan, and violet lights ripple across the glossy street like liquid electricity, evoking a futuristic vision of the United Arab Emirates as a nexus of technology, mystery, and urban energy under a dark, rain-streaked sky.

- **Sample-12**. A cybernetic warrior stands resolute in the heart of a rain-lashed, neon-soaked metropolis, his face etched with intricate biomechanical tattoos that glow faintly under the pulsating pink and blue lights of towering holographic billboards, while his eyes are hidden behind sleek, futuristic visor goggles radiating a cool violet-blue luminescence that mirrors the city's electric pulse; clad in a high-collared, armored black jacket accented with glowing orange circuitry along its seams, he exudes an aura of stoic intensity and technological prowess, as blurred silhouettes of passersby dissolve into the background, their forms swallowed by the misty haze and shimmering reflections on wet pavement, immersing him in a world where humanity and machine merge beneath the ceaseless drizzle and chromatic glow of a dystopian urban dreamscape.

- **Sample-13**. As the sun dips below the horizon, casting a warm golden glow across the sky that fades into soft blues and purples, Shanghai's iconic Oriental Pearl Tower stands tall and radiant, its spherical sections glowing with pink and purple hues that mirror the twilight, anchoring the city's futuristic skyline against a backdrop of sleek glass skyscrapers and modern high-rises; below, the Huangpu River flows gently, reflecting the fading light and the silhouettes of bridges and riverside trees, while lush green foliage along the embankment frames the scene, adding a touch of nature to the urban grandeur, creating a serene yet dynamic panorama where technological marvels and natural beauty converge in perfect harmony at dusk.

- **Sample-14**. Under a brooding, leaden sky that looms heavy with the promise of storm, a colossal wave rises in furious majesty—its dark, churning body sculpted by unseen winds into a towering, curling crest that crashes forward in a froth of white foam and spray, its deep indigo and slate-gray depths hinting at the ocean's raw, untamed power; above the tumult, a scattered flock of seabirds soars with outstretched wings, their silhouettes stark against the gloom as they ride the turbulent air currents, embodying both freedom and resilience amid nature's overwhelming force, while the horizon vanishes beneath the swell, leaving only the primal drama of sea and sky locked in eternal, awe-inspiring conflict.

- **Sample-15**. The image showcases a delectable pepperoni pizza presented on a rustic wooden board, set against a dark, textured background that adds a touch of sophistication. The pizza boasts a golden-brown crust with visible char marks from being cooked in a wood-fired oven, indicating a crispy texture. The cheese, melted and slightly browned in spots, blankets the pizza evenly, with some areas showcasing a rich, gooey appearance. The toppings are predominantly pepperoni slices, arranged in a somewhat circular pattern around the edges, while others lie scattered across the surface in various orientations. Each slice of pepperoni is glossy, indicating a fresh, juicy texture, and they are generously placed, making the pizza look hearty and appetizing. Interspersed among the pepperoni slices are small flecks of herbs, likely basil, adding a burst of green color and freshness to the dish. To the right side of the pizza, two fresh basil leaves are artistically placed, their vibrant green hues contrasting beautifully against the warm tones of the pizza and the wooden board. A few more basil leaves can be seen in the foreground at the bottom left corner, scattered more casually than the ones on the pizza itself. There are also a couple of slices of pepperoni lying outside the pizza, further enhancing the visual appeal of the presentation. The overall composition of the image is balanced, with the pizza centrally located, drawing the viewer's attention immediately. The lighting is subtle yet adequate to highlight the textures and colors of the pizza, making it look inviting and mouth-watering. The slight shadows cast by the pizza and basil leaves add depth to the image, creating a three-dimensional feel.

- **Sample-16**. The image depicts a serene night scene at a lively port town. The sky is filled with a bright starry Milky Way galaxy, casting a soft glow over the entire scene. The town features quaint, charming houses with warm yellow lights emanating from their windows, creating a cozy ambiance. At the forefront, there is a group of people gathered around wooden tables, enjoying their time together. They are engaged in conversation and laughter, with cups of coffee or tea in hand. A golden retriever dog sits by one of the tables, adding to the homely atmosphere. To the right, there is a tall streetlight and a small flower arrangement in a pot, further enhancing the quaint

charm of the setting. In the background, a harbor is visible with boats anchored, and the town extends with more houses and shops lining the streets, including a bakery sign.

- **Sample-17**. From a high vantage point, the sun rises—or sets—in a blaze of golden-orange light that pierces through a dramatic sky streaked with soft pink, lavender, and deep blue clouds, casting long, ethereal shadows across a vast, snow-blanketed landscape of rolling hills and undulating valleys where a winding road snakes like a ribbon through the serene white expanse; frost-kissed shrubs dot the foreground, their dark branches dusted with snow and catching the warm glow, while the distant horizon fades into a hazy, dreamlike mist, blending earth and sky in a tranquil, almost otherworldly winter tableau that evokes both solitude and sublime beauty beneath the celestial spectacle of dawn or dusk.

- **Sample-18**. The image captures the majestic Forbidden City in Beijing, China, bathed in the warm hues of a setting sun. The scene is dominated by several large, traditional Chinese buildings with elegant, ornate roofs painted in vibrant reds and golds. These buildings feature numerous golden dragons and intricate carvings, typical of imperial architecture. The main structure in the center is an imposing palace with multiple eaves and large golden pillars, its entrance flanked by smaller pavilions. The central building's roof is adorned with intricate patterns and two large, pointed gables, adding to its grandeur. In front of the palace, a wide, open courtyard stretches out, paved with smooth, light-colored stones and bordered by white stone balustrades. These balustrades are decorated with sculpted figures and floral designs, providing a stark contrast to the dark stone of the buildings behind them. The courtyard is devoid of people, emphasizing the serene and historical atmosphere of the site. To the left, more buildings can be seen, each with their own distinct architectural features, though slightly obscured due to the architectural layout. The sky above is a soft gradient from pale blue at the horizon to a warm orange near the sun, which casts a gentle glow over the entire scene. A few wispy clouds are scattered across the sky, adding depth and dimension to the panoramic view. In the foreground, there is a series of white, stone railings and steps leading up to the palace, guiding the viewer's eye towards the impressive structure. The entire area is bathed in the soft, golden light of the sunset, creating a peaceful and timeless quality that highlights the historical significance of this famous landmark.

- **Sample-19**. In this serene, sunset-hued beach scene, a woman stands with her back to the viewer, gazing out at the ocean. She has long brown hair tied loosely behind her head and wears a flowing white sleeveless dress that reaches her ankles. She carries a pair of black flip-flops in her right hand. Her light brown and white dog sits attentively beside her on the sandy shore, their brown and white fur contrasting with the warm, golden tones of the setting sun. The beach is bathed in the soft, orange glow of the setting sun, casting long shadows and highlighting the texture of the sand. In the distance, the gentle waves roll onto the shore, with the sun's reflection shimmering on the water. To the left, a sailboat sails across the calm sea, its silhouette silhouetted against the warm sky. A wooden lifeguard chair with a red life buoy stands near the center-right of the scene, next to a blanket with a floral pattern draped over its legs. The beach is dotted with footprints, and tall grasses and shrubs frame the scene. A couple of seagulls fly low in the orange sky, adding to the tranquil atmosphere. In the background, a cliff rises, partially obscuring the view, and a few more sailboats are visible on the horizon.

- **Sample-20**. An ancient Greek philosopher is talking on a wireless headset.

- **Sample-21**. A serene elven woman with pointed ears and intricate silver face art gazes thoughtfully, clad in a dark green gown with gold trim. She stands in a mystical, moonlit forest where glowing blue mushrooms illuminate the shadowy trees around her.

- **Sample-22**. The image depicts a small, well-lit home office setup in a cozy room with beige carpeting. The primary focus is a compact wooden desk positioned against a pale wall. The desk has a simple, light-colored finish and is supported by two metal legs, which appear to be adjustable for height. On the desk, there is a black keyboard and a laptop computer on the right side, along with a closed, black-framed flat-screen monitor to the left of the laptop. A white mouse and a pair of sunglasses rest on the keyboard. A single table lamp with a black shade stands next to the keyboard, casting a warm light over the workspace. To the left of the lamp, a small stack of books or papers rests on the desk surface. A black rolling chair with height-adjustable arms is stationed directly in front of the desk. The chair's wheels are visible, indicating its portability. The computer monitor is accompanied by a webcam mounted above it on the wall. Below the desk, the floor is partially covered with a light-colored rug that contrasts with the carpeting. Adjacent to the desk, there is a potted plant with lush green leaves placed on a small round table or stand. The room's background features a bookshelf filled with various books, some of which are visible through open shelves. A white cushioned armchair sits to the left of the desk, suggesting a cozy

nook for relaxation or additional seating. On the wall behind the desk, near the corner, a rectangular mirror reflects part of the room, adding depth to the space. An overhead lighting fixture casts a soft yellow glow from above, highlighting the desk area while keeping the rest of the room dimly lit. The overall color palette includes neutral tones—beige, white, and shades of brown—creating a calming and functional workspace atmosphere.

- **Sample-23**. In a vibrant, arid desert landscape bathed in warm, golden hues of sunset, a group of three individuals ventures through a rugged, canyon-like terrain. The woman at the center, dressed in a practical olive-green safari outfit with rolled-up sleeves, khaki pants, and a belt bag slung over her shoulder, walks confidently towards the camera. Her dark hair is tied up in a bun, and she has a focused expression on her face as she gazes at the ground. A small, playful fox stands beside her, attentively looking ahead. The woman's right hand holds a stainless steel water bottle, and her left arm is relaxed by her side. On the right, a man wearing a wide-brimmed straw hat, beige shirt, and cargo pants stands observing the surroundings, while his young son, dressed in an orange t-shirt and black shorts, looks back at them with a curious expression. The man and his son are positioned slightly behind the woman, who appears to be leading the way. In the foreground, a cactus plant with a yellow bloom adds to the desert ambiance. The background features towering red rock formations and sparse vegetation, including a few Joshua trees and desert scrub. A large eagle soars high above, its wings spread wide against the backdrop of a sky painted with swirling clouds in shades of orange, pink, and purple. The sand beneath their feet is dotted with footprints, suggesting they have been walking for some time. The entire scene is imbued with a sense of adventure and exploration, set against the timeless beauty of a desert canyon under a dramatic sunset sky.

- **Sample-24**. Please generate a realistic image of the traditional Chinese Hotan Jade pendant. The pendant is a round jade brand, with a full color of turquoise. The jade is warm and delicate, and the surface is highly polished but not excessively reflective, showing the oily texture of real jade. A traditional Jiangnan garden landscape painting is carved in relief on the jade plaque: the upper half of the picture shows a group of Chinese style buildings arranged in a staggered manner, with roofs featuring upturned eaves and horsehead walls, and rich details. The buildings are interspersed with delicate elements such as small bridges, flowing water, weeping willows, and rockeries. The overall composition is complex but not messy, presenting a freehand feeling of traditional Chinese painting style. The lower part of the screen is relatively blank, with only winding rivers flowing from bottom to right, enhancing the spatial hierarchy. The pendant is hung on a gray green Chinese woven rope, with a simple and natural knot, tightly woven from multiple strands of fine thread, with a tough texture. It is decorated with a small red coral bead directly above it. The background of the picture is a light gray cloth in the style of physical photography as a reference, which is overall realistic and realistic. The style is modern high-quality still life photography, with clear composition, soft lighting, and focus on the center of the jade plaque, blurring the background details.

- **Sample-25**. A portrait of profound wisdom and quiet contemplation, this elderly man with a long, flowing white beard and deeply lined face is captured in dramatic chiaroscuro lighting against a dark void, his gaze fixed on something unseen beyond the frame.

**More "Image-Text-Image" reconstruction results by our method.** Here, we provide more visual examples of the "image-text-image" pipeline using our method, i.e., our encoder processes the input image, generate the output descriptive caption, and then pass it through our decoder to recover it to pixel. See Fig. 13, Fig. 14, and Fig. 15 for details.

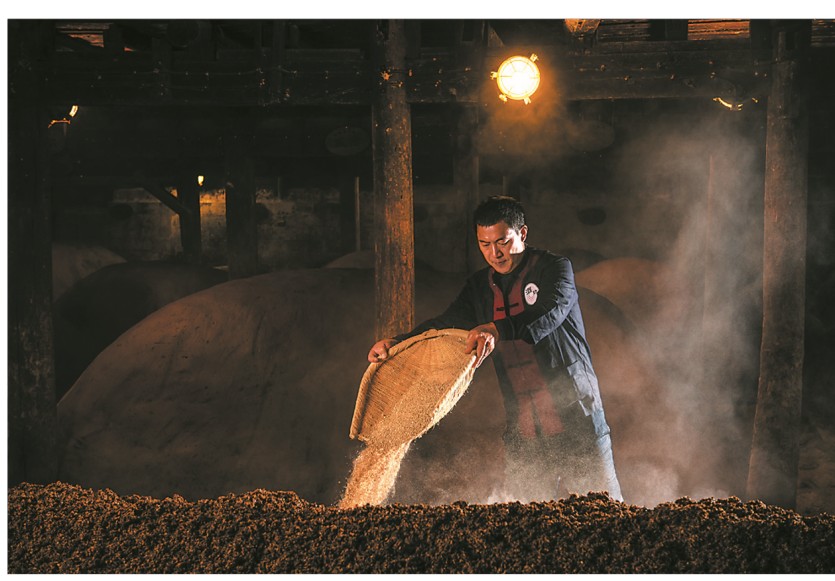

This is an indoor photograph depicting a man engaged in traditional grain processing, creating a warm and historically rich atmosphere.

### General Overview
In the photo, a man is winnowing grain with a tray, set against a dimly lit indoor background with soft lighting, evoking an ancient and focused ambiance.

### Main Subject Identification
- **Core Subject**: A man positioned slightly to the right of the center.
- **Ethnicity and Age**: The man appears to be of Asian descent, aged between 30 and 40.
- **Expression**: His expression is concentrated, with his gaze fixed on the grain in the tray.

### Detail Characteristics
- **Attire**: The man is wearing a dark jacket over a red garment, with a circular badge on the jacket.
- **Hair**: Short and neatly styled.
- **Tray**: He holds a light-colored bamboo tray with both hands, which is semi-circular and made of natural bamboo.
- **Grain**: The grain in the tray is light yellow, being tossed to form an arc towards the ground.

### Quantity and Position
- **Number of Subjects**: One person.
- **Position**: The man is slightly to the right of the center, with the tray between his hands, and grain being scattered onto a pile on the ground.
- **Background**: There are several large grain piles in the background, dome-shaped, located behind and to the left of the man.

### Actions and Background
- **Action**: The man holds the tray with both hands, slightly leaning forward, with his right hand on the right side of the tray and his left on the left, winnowing the grain. His legs are slightly apart for balance.
- **Background Scene**: The setting is a wooden indoor structure with dim lighting. A bright bulb hangs from the ceiling, casting a warm yellow glow. The air is filled with grain dust, adding dynamism to the scene.

### Artistic Style and Techniques
- **Style**: Realistic, emphasizing detail and texture.
- **Lighting Effects**: Light from above creates a Tyndall effect, making the grain dust visible and enhancing the depth of the image.
- **Framing and Perspective**: Medium shot, centered composition, with the camera angle slightly below the man's eye level, making the subject more prominent and lifelike.

### Language Standards
- **Objective Description**: Only visible content is described, without speculation or irrelevant information.

### Conciseness
- **Precise Description**: Covers all core information, avoiding redundancy. This detailed description comprehensively captures the scene and atmosphere of the photograph.

Figure 9: Visual example of the proposed 700k long-context text-to-image dataset.

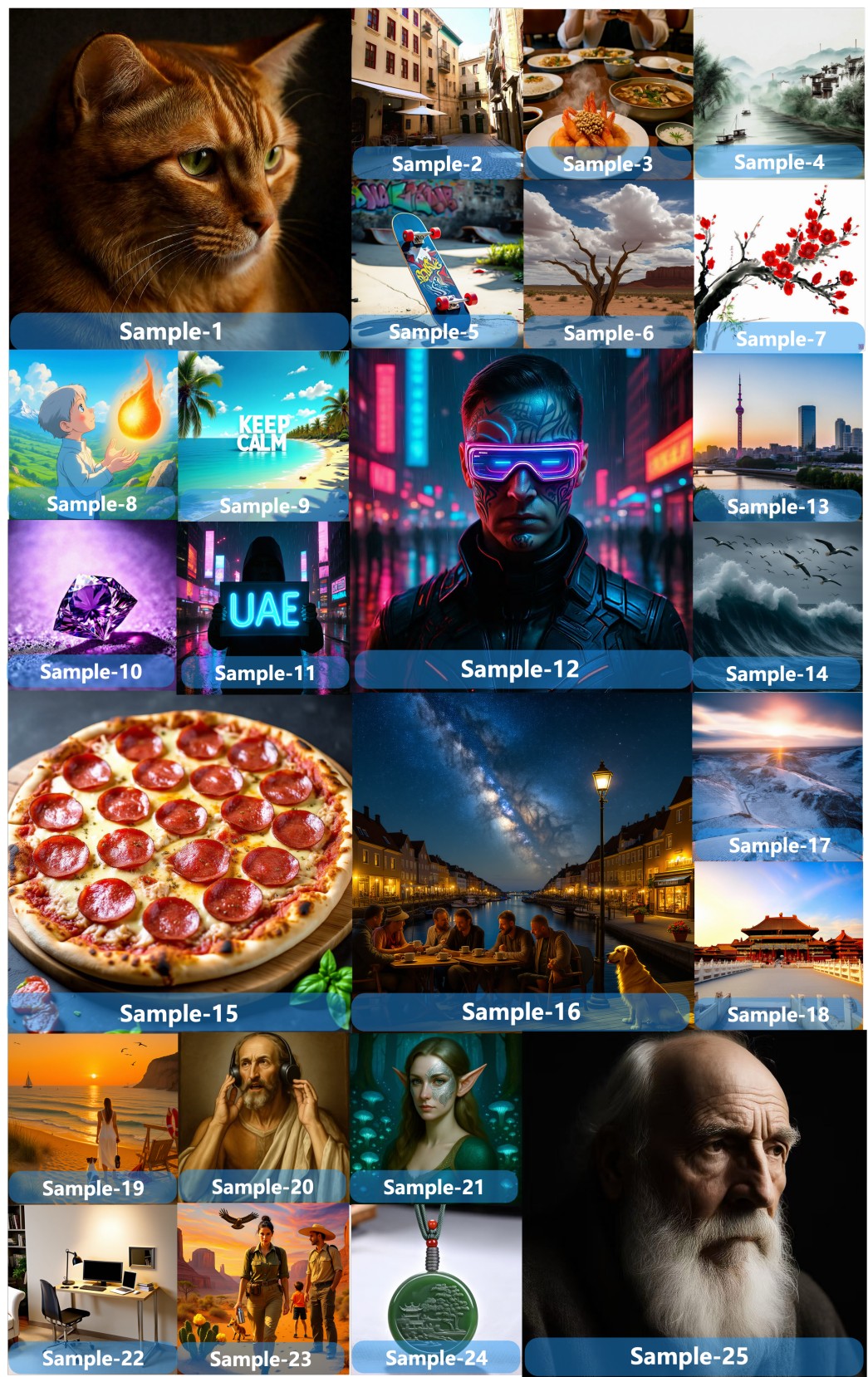

Figure 10: Visualization results of UAE at 1024×1024 resolution.

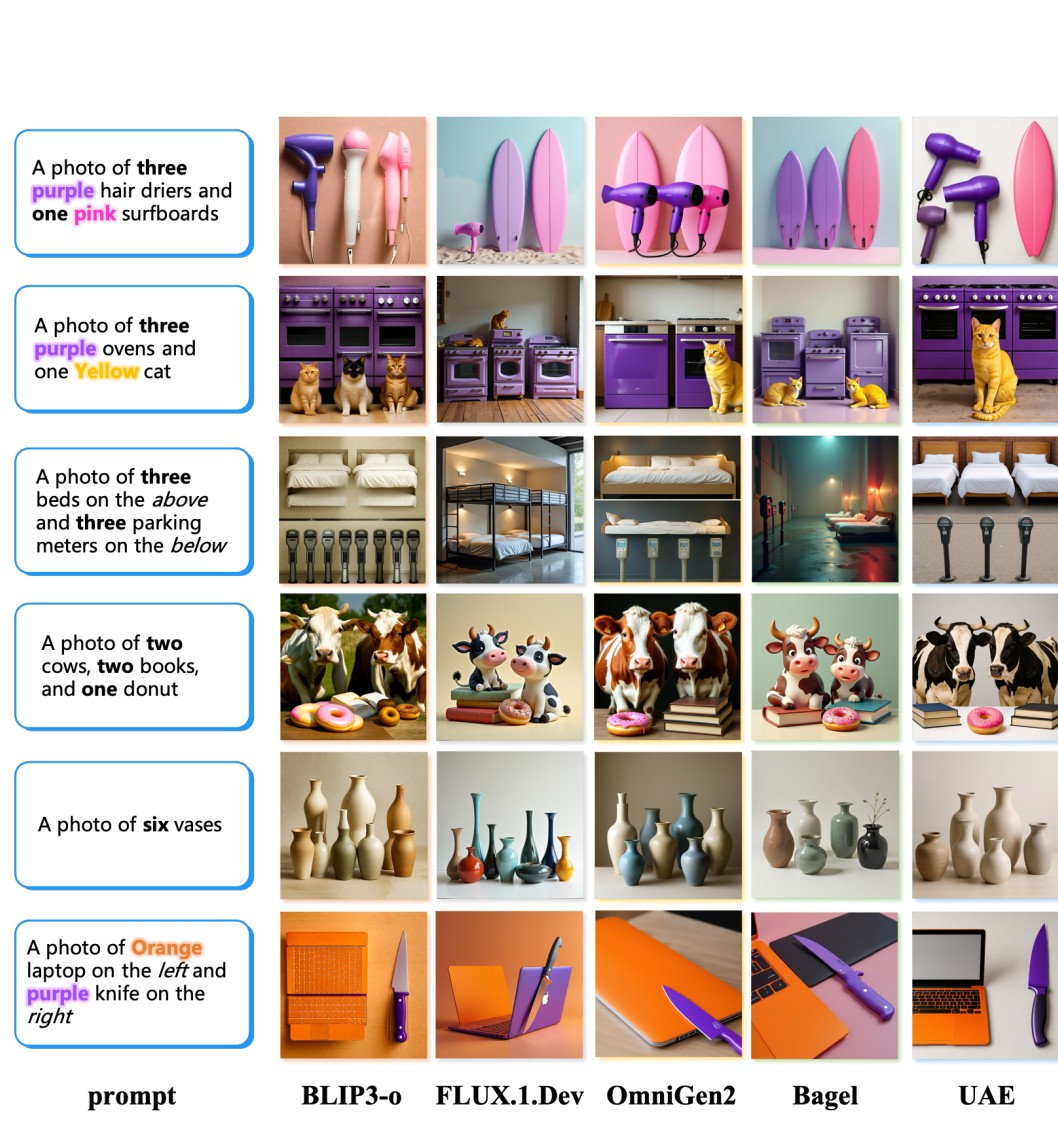

Figure 11: Qualitative analysis of the results from GenEval++, where our UAE demonstrates visually consistent results aligned with the input captions, and performs reasonable composition for each element.

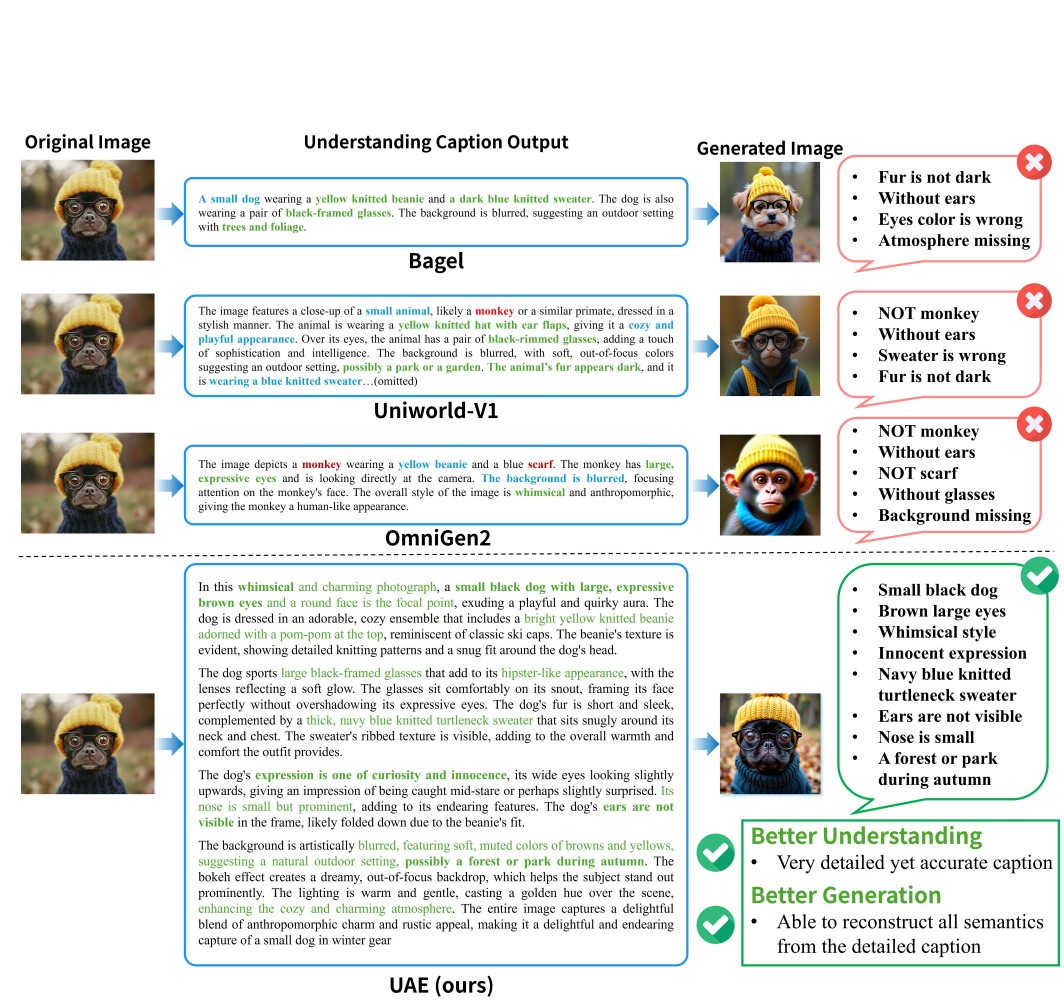

Figure 12: Case study of the results from the proposed Unified-Bench, we see that our UAE enables to produce a more detailed, accurate, comprehensive description based on the input image, and reconstructs a similar result to the original image, showcasing the improved understanding and generation capabilities, and the better unification of the system.

**Original
Image**

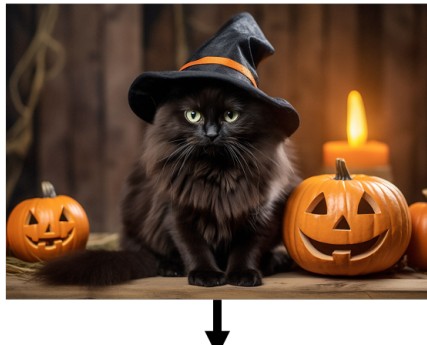

**Generated
Caption

(by Encoder)**

In this Halloween-themed image, a **black cat** with striking **green eyes** sits gracefully on a wooden surface adorned with a rustic ambiance. The cat is **wearing a classic witch's hat**, characterized by its pointed tip and **soft black fabric** with an orange trim along the brim. This hat rests elegantly atop the cat's head, adding to its mystical appearance.

The **cat's fur is long and dense**, giving it a fluffy texture that contrasts beautifully with the matte finish of the hat. Its white whiskers stand out prominently against its dark fur. The cat's **expression is calm and composed**, looking directly at the camera with a gentle gaze.

Surrounding the cat are **three carved pumpkins**, each featuring a traditional jack-o'-lantern face with triangular eyes, a wide smile, and a jagged mouth. The closest pumpkin to the cat has a prominent flame rising from within, casting a warm, flickering light that adds to the festive atmosphere. **The other two pumpkins are placed slightly further away, one to the left and another to the right**.

The background is blurred but reveals a cozy, dimly lit room with a wooden wall, which enhances the warm, cozy feel of the scene. **A lit candle with a flickering flame** is positioned near the right side of the frame, contributing to the Halloween theme. The overall composition of the image combines elements of Halloween tradition with a whimsical twist using the cat, making it both festive and charming.

**Generated
Image

(by Decoder)**

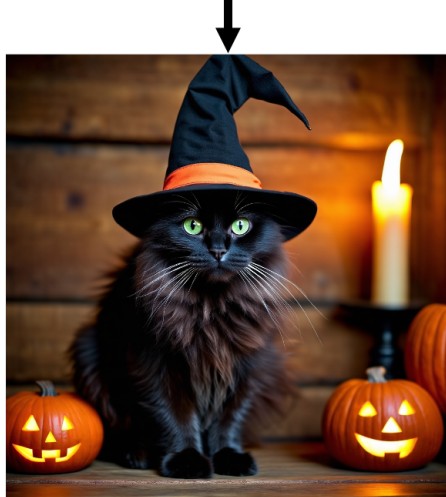

Figure 13: Example of the "image-text-image" reconstruction result by **our encoder and decoder**.

**Original Image**

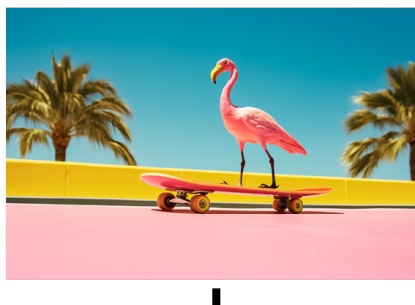

**Generated Caption**

**(by Encoder)**

In this vibrant and surreal photograph, a pink flamingo **stands atop a pink skateboard**, defying gravity as if taking flight on a miniature skate park. The flamingo, with its elongated neck, **bright yellow beak**, and delicate legs, appears almost lifelike, rendered in pastel shades of pink and white feathers that shimmer slightly under the sun. Its stance is confident.

The skateboard itself is a vivid pink, featuring **yellow wheels—front** and back—with intricate detailing that adds to its playful appearance. The board rests on a **pink surface**, which could be misunderstood as pavement or ground but is more likely a painted platform due to the uniform color and matte texture. This pink area extends across the lower half of the image, providing a striking contrast to the clear, bright pink shadow cast by the flamingo onto the ground.

In the background, the scene unfolds against a vivid blue sky, suggesting a sunny day. **Two palm trees**, their leaves shimmering in the sunlight, frame the image, **one on the left and another on the right**, both slightly out of focus, emphasizing the **flamingo as the central subject**. The yellow wall behind the flamingo is short and segmented, adding a pop of contrasting color to the otherwise monochromatic backdrop. The wall's yellow hue matches the skateboard, creating a harmonious visual connection between the elements. The entire setting exudes a **whimsical, retro vibe reminiscent** of vintage skateboarding culture, with the flamingo adding an unexpected twist to the classic scene.

**Generated Image**

**(by Decoder)**

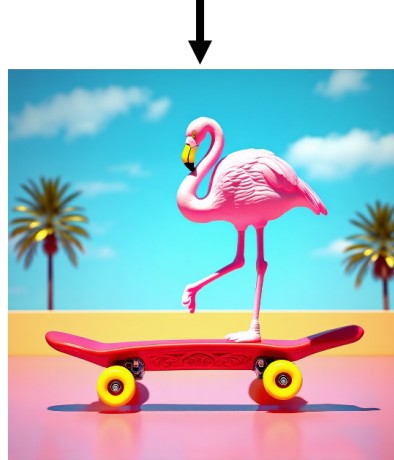

Figure 14: Example of the "image-text-image" reconstruction result by **our encoder and decoder**.

**Original Image**

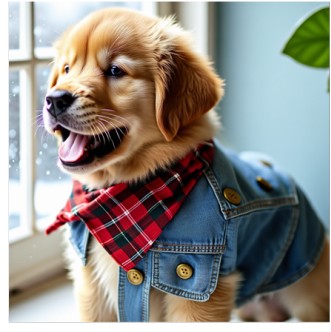

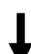

**Generated Caption**

**(by Encoder)**

In this heartwarming photograph, a **golden retriever puppy** stands with a joyful expression, **gazing slightly to its left**. Its fur is a golden-brown, well-groomed and shiny, with a soft, fluffy texture. The puppy has **small, dark eyes** that sparkle with excitement and a friendly demeanor. Its ears are floppy and slightly darker at the tips, framing its face well.

The puppy wears a **red plaid scarf** that sits snugly around its neck, featuring traditional tartan patterns in shades of **red, black, and white**. The scarf is complemented by a denim jacket with a denim collar, giving it a cozy, winter look. The denim jacket has a casual yet stylish feel, showcasing thick, medium-blue denim fabric with visible stitching and golden metal buttons and snaps. The puppy sports a big, round collar bone on its chest and its **denim jacket features two pockets with matching gold buttons**.

The **puppy's mouth is open**, and its tongue protrudes gently, adding to its adorable charm. Its black nose contrasts sharply against the light fur of its face, and it has a small, dark nose ring just above it.

**The puppy stands indoors near a window**, bathed in soft, natural light filtering through, creating a warm and cozy atmosphere. Behind the puppy, a window reveals a blurred, sparkling background likely containing snowflakes or blurred lights, suggesting a winter scene. The window itself has white wooden frames, partially obscuring the light. **To the right, a light blue wall provides a soft backdrop**, while the puppy's denim jacket adds a textured element to the image.

Behind the dog, a **blurred green plant leaf peeks into the frame**, contributing to the indoor ambiance. The overall composition is balanced and inviting, emphasizing the puppy's cheerful spirit and the cozy, festive environment it inhabits.

**Generated Image**

**(by Decoder)**

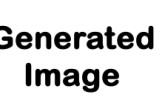
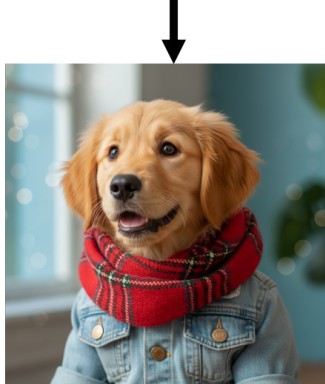

Figure 15: Example of the "image-text-image" reconstruction result by **our encoder and decoder**.

## F    THE USAGE OF LARGE LANGUAGE MODELS

In this paper, large language models (LLMs), like ChatGPT, are used only for writing refinement and grammar correctness. We do not use it for idea proposal, research design, data analysis, or interpretation of results.

## G    REPRODUCIBILITY STATEMENT

To ensure the reproducibility of this work, we have made significant efforts to provide comprehensive details of our methodology and experiments (see Appendix Sec. C for details). To facilitate the reproduce, we promise we will make our proposed dataset, benchmark, complete source code, pre-trained model weights, and experiment configurations publicly available upon publication.

