# OpenReview forum: "Unified Multimodal Model as Auto-Encoder"
_ICLR.cc/2026/Conference — ICLR 2026 Conference Withdrawn Submission_

### Official Review · Reviewer_M6H5 · 2025-10-23

**Soundness:** 2
**Presentation:** 3
**Contribution:** 2
**Rating:** 2
**Confidence:** 4

**Summary:**

This paper proposes UAE, a framework that attempts to unify multimodal understanding (I2T) and generation (T2I) under an auto-encoder perspective. The encoder (Qwen2.5-VL) converts images into descriptive captions, while the decoder (SD3.5) reconstructs images from those captions. The authors introduce Unified-GRPO, a reinforcement learning scheme optimizing a unified reconstruction objective, and present Unified-Bench to measure unification via image reconstruction similarity. Experiments on GenEval, GenEval++, and MMT-Bench suggest mutual benefits between understanding and generation.

**Strengths:**

The paper aims to address an important challenge in achieving genuine unification between multimodal understanding and generation. The auto-encoder analogy is intuitive, offering a novel perspective on cross-modal consistency. Unified-GRPO introduces an RL-based training pipeline, while Unified-Bench serves as an evaluation framework for reconstruction-based multimodal systems. The experimental section is extensive, covering multiple generation benchmarks and perceptual analyses.

**Weaknesses:**

- UAE is essentially a composition of two fully independent pretrained models: Qwen2.5-VL (encoder) and SD3.5 (decoder). There is no shared latent space, joint training, or parameter-level integration between them. As a result, the claimed unification is achieved solely through a reconstruction loss, rather than in the model representation.

- The unified reconstruction objective and Unified-GRPO reward are defined by image reconstruction fidelity. The encoder is optimized solely to assist the decoder in producing better reconstructions, which diminishes the emphasis on visual understanding itself. As a result, while the framework is presented as a unified multimodal model (UMM), it primarily focuses on generation rather than generalizable and comprehensive multimodal reasoning or comprehension.

- Although the paper frames UAE as an auto-encoder, it still requires an additional large-scale paired dataset (LongCap-700k) despite the use of the pretrained encoder and decoder. This heavy reliance on external paired supervision undermines the conceptual claim of self-contained unification.

- Using natural language captions as the intermediate representation introduces an irreversible information bottleneck. Language struggles to capture fine-grained visual details, such as texture, geometry or color continuity, making the concept of a "near-lossless bidirectional flow" theoretically implausible.

- The experiments heavily focus on generation benchmarks (GenEval, GenEval++), while the understanding capability is only evaluated on MMT-Bench. As a unified multimodal model (UMM), visual understanding tasks should be evaluated on a broader range of benchmarks, including POPE, GQA, MMMU, and others.

**Questions:**

Please see weaknesses.

---

### Official Review · Reviewer_d9E4 · 2025-10-30

**Soundness:** 2
**Presentation:** 2
**Contribution:** 2
**Rating:** 2
**Confidence:** 4

**Summary:**

This paper aims to address the schism between multimodal understanding and generation in unified models. It proposes the UAE framework, inspired by Auto-Encoder: treating understanding as an I2T encoder and generation as a T2I decoder, with high-fidelity reconstruction as the core objective. The proposed Unified-Bench evaluates unification. Experiments show UAE boosts generation (e.g., GenEval) and strengthens fine-grained perception (e.g., MMT-Bench), proving mutual benefits between understanding and generation in unified models.

**Strengths:**

This paper formulates the two tasks of the unified model within an autoencoder paradigm, which is a conceptually sound approach. The experimental setup is sufficiently thorough and well-reasoned. Additionally, the paper is well-written and demonstrates a substantial amount of work.

**Weaknesses:**

1. The authors present many of their contributions as “the first work,” but they are actually not new. Treating I2T and T2I as two parts of an autoencoder is not a new idea; papers such as De-Diffusion [1] proposed this concept long ago. Second, the authors claim that unified-GRPO is the first RL algorithm to implement mutual bons; however, both [2], UniRL [3], and ULM-R1 [4] have already implemented the mutual bon of a unified model. Using these concepts or continuing to improve upon them is not a bad thing or a problem; a valuable contribution is not determined by whether it is the first work. What’s needed is an honest discussion of the strengths and weaknesses of one’s work, and avoiding overstatement, which can be misleading to people outside the unified model field.

2. The “generation for understanding” claim is quite odd. This is essentially an image captioning task, and the authors somewhat overclaim its value. The reason for the “text increasingly longer” phenomenon is that you trained the decoder on a long-caption dataset. The training data for generation are image–prompt pairs; by doing so, you are fitting the prompt distribution of the training data. Prompts/captions from the same distribution will naturally yield generated images that are more similar to the original images. I suggest not using long captions for pretraining, or directly comparing the average length of the increasingly longer text with the average length in longcap-700k. Otherwise, this training setup is merely using the encoder to fit the decoder’s training data distribution and cannot substantiate “generation for understanding.”

3. In addition, I suggest the authors experimentally verify whether, on the generation side, directly using CLIP or other similarity rewards between the input image and the output text (caption) to perform GRPO would also exhibit the “text increasingly longer” phenomenon. The image modality is dense while text is sparse; in principle, such experiments would also lead to “text increasingly longer.”

4. To demonstrate the generality of the method, I recommend conducting experiments across models with different architectures, such as janus-pro, bagel, and showo2.

[1] De-Diffusion Makes Text a Strong Cross-Modal Interface

[2] Turning Internal Gap into Self-Improvement: Promoting the Generation-Understanding Unification in MLLMs

[3] UniRL: Self-Improving Unified Multimodal Models via Supervised and Reinforcement Learning

[4] Co-Reinforcement Learning for Unified Multimodal Understanding and Generation

**Questions:**

I do not think this paper is currently at the level for acceptance at ICLR. I hope the authors can address the questions I outlined in the Weakness section, after which I will reconsider my score.

---

### Official Review · Reviewer_fMrL · 2025-10-30

**Soundness:** 3
**Presentation:** 2
**Contribution:** 4
**Rating:** 8
**Confidence:** 3

**Summary:**

This paper proposes UAE (Unified Auto-Encoder), a novel framework for unifying multimodal understanding and generation. The authors (1) introduce a LongCap-700k dataset containing long, descriptive captions for fine-grained text-to-image training; (2) pretrain a diffusion-based decoder (DiT) conditioned on large vision-language model (LVLM) embeddings; and (3) propose Unified-GRPO, a reinforcement learning scheme that alternates between “Generation for Understanding” and “Understanding for Generation” phases to jointly optimize both models for reconstruction consistency. They also introduce Unified-Bench, a new benchmark measuring bidirectional coherence between modalities. Experiments show the effectiveness of the proposed method.

**Strengths:**

1. Novel post-training strategy: The design of Unified-GRPO is interesting and it significantly increases the performance.

2. New benchmark: They introduce Unified-Bench, a new benchmark measuring bidirectional coherence between modalities.

3. Great experimental results. Experiments show state-of-the-art results on GenEval, GenEval++, and MMT-Bench, as well as strong unified scores compared to GPT-4o-Image and OmniGen2.

4. Comprehensive evaluation: Unified-Bench is thoughtfully designed to assess cross-modal coherence beyond conventional metrics.

5. Reproducibility: The authors promise to release the dataset, benchmark, source codes and model weights. These resources are valuable.

6. Clarity: The paper reads smooth and mostly clear.

**Weaknesses:**

1. Reward design choices: The proposed UnifiedGRPO is the core contribution. More ablation studies and analysis required. For example, the reconstruction similarity is used as a unified reward, but the paper lacks ablation on alternative reward formulations (e.g., CLIPScore vs. perceptual vs. DINO metrics).

2. Missing discussion with related works. (1) It is suggested to compare and discuss with OpenUni [a], a simple yet strong baseline for unified LMMs. OpenUni also has a similar model size (using InternVL3-2B as the backbone), and achieves 0.86 on GenEval. (2) In addition, FlowGRPO achieves 0.95 on GenEval, using RL training strategy which is not reported in Table 2. The comparisons with FlowGRPO is critical as they also adopts RL. (3) RecA [b] also presents a similar idea i.e. reconstruction alignment. It is suggested to  add some discussions.

3. Repeated citations. In line 738-742.

[a] Wu S, Wu Z, Gong Z, et al. OpenUni: A Simple Baseline for Unified Multimodal Understanding and Generation[J]. arXiv preprint arXiv:2505.23661, 2025.
[b] Xie J, Darrell T, Zettlemoyer L, et al. Reconstruction alignment improves unified multimodal models[J]. arXiv preprint arXiv:2509.07295, 2025.

**Questions:**

1. How sensitive is Unified-GRPO to the choice of reward metric and group size G?

2. Could the decline in text/OCR performance be mitigated by hybrid symbolic-textual reconstruction losses?

---

### Official Review · Reviewer_N9UK · 2025-10-30

**Soundness:** 3
**Presentation:** 3
**Contribution:** 3
**Rating:** 2
**Confidence:** 3

**Summary:**

This paper introduces the Unified Multimodal Model as Auto-Encoder (UAE) framework, proposing a novel paradigm to resolve the long-standing schism between multimodal understanding (Image-to-Text encoder) and generation (Text-to-Image decoder). The core idea is to unify these two tasks under a single objective: high-fidelity reconstruction, utilizing a LongCap dataset for pre-training and a reinforcement learning-based strategy, Unified-GRPO, to foster mutual enhancement between the encoder and decoder modules.

**Strengths:**

- Novel Unification Paradigm: The conceptual shift to viewing I2T and T2I as an Auto-Encoder with reconstruction fidelity as the single, unified objective is theoretically elegant and highly novel in the context of UMMs.
- Methodological Innovation: The proposed Unified-GRPO strategy, utilizing RL for joint optimization and mutual benefit, is a creative approach to fine-tuning multimodal models.
- Empirical Strength: The model (UAE) achieves competitive performance on the newly proposed Unified-Bench benchmark, suggesting a high degree of integration between the two modalities.

**Weaknesses:**

1. Generalizability and Scope of the UAE Paradigm (Major Comment)
The UAE paradigm achieves mutual enhancement through the joint training of distinct I2T Encoder and T2I Decoder modules. This decomposition into dedicated components inherently raises concerns about its practical scope and generalizability. Specifically:
Does this two-module architecture inherently restrict the applicability of UAE?
Can monolithic UMMs or existing unified models (e.g., BAGEL[1], Janus-pro[2], Show-o[3]) adopt or adapt the UAE training principles (especially the Unified-GRPO strategy) to enhance their performance, or is UAE strictly limited to models that maintain separate encoder/decoder roles? The authors should discuss the feasibility of applying the core philosophy of "reconstruction as a unified objective" to existing, non-AE model architectures.

2. Limitations and Catastrophic Forgetting from Decoder's Constraints (Critical Comment)
The "Generation for Understanding" stage optimizes the Encoder based on the reconstructibility score provided by the Decoder. This mechanism implies that the Encoder will learn to omit information that the Decoder cannot accurately reproduce, potentially leading to catastrophic forgetting or degradation in specific fine-grained understanding capabilities.
The authors briefly mention this phenomenon regarding text content and OCR tasks (Line 468-469), stating that the current diffusion-based Decoder struggles to render text accurately. This limitation is highly critical as it highlights a fundamental flaw: the Encoder's "understanding" is capped by the Decoder's "generation fidelity."
The paper must provide a much richer and more elaborate discussion on this trade-off. Specifically, the discussion should cover:
Impact on Other Fine-Grained Data Types: Beyond OCR, how does this limitation affect the Encoder's ability to retain knowledge about other complex, hard-to-render visual details, such as subtle textures, specific material properties, or numerical data (e.g., small chart data)?
Decoder Agnosticism: The authors should discuss how the UAE paradigm would perform if different types of Decoders were used (e.g., Flow-based models, specialized rendering networks, or VQ-VAE based decoders), and whether these alternative Decoders could mitigate the loss of OCR/fine-grained information.
Mitigation Strategies: What explicit mechanisms (e.g., multi-objective loss, auxiliary losses for specific tasks) can be integrated into the Unified-GRPO framework to prevent the Encoder from discarding these critical, non-reconstructible details, thus ensuring the "unified" model remains strong in all modalities?

3. Fairness of Protocol 1 Comparison
I have concerns about the comparison methodology in Protocol 1:
Lines 179 mentions the Decoder uses the Encoder's mapped last hidden state, while Lines 251-252 states that other baseline models use the standard text caption. This discrepancy in the intermediate representation used for T2I generation appears to constitute an unfair comparison, as one is a continuous embedding and the other is discrete text.
Even if the UAE model could be tested using the text caption in Protocol 1, the test itself measures the similarity of the reconstructed image to the original, which is the UAE model's primary training objective. Therefore, the high performance of UAE in Protocol 1 is structurally advantageous to its own framework, significantly reducing the protocol's reference value as a neutral measure of unification compared to baselines. The authors should consider mitigating this bias or focusing the interpretation of Protocol 1 results more narrowly.

4. Clarity in Figure 2 and Model Representation
The visualization in Figure 2 requires clarification to prevent reader confusion:
In Stage 1, the flow is depicted as "Image -> (frozen) LVLM -> text." This LVLM appears to be a standard, non-finetuned model used for initial caption generation, which is distinct from the LVLM (Encoder) refined in Stage 2 and 3. This lack of differentiation in the figure's label could mislead readers into thinking all stages use the same LVLM configuration.
Line 179 states that the Decoder uses the mapped last hidden state of the Encoder. However, Stage 2 and Stage 3 of Figure 2 clearly show the intermediate representation as "text." The authors must reconcile this apparent contradiction and clearly articulate whether the reconstruction process during training (Unified-GRPO) uses: (a) discrete generated text, (b) continuous hidden states, or (c) a combination of both. A clear visual representation in Figure 2 is crucial.

5. Missing Evaluation on Reasoning Benchmarks
The paper strongly advocates for the notion of "Understanding for Generation" and the mutual benefit of the two tasks. However, the evaluation seems to omit standard benchmarks designed to test the reasoning and complex instruction following capabilities of unified models in generative tasks.
Benchmarks that evaluate the generative model's inference capacity, such as WISE[4], should be included. Reporting performance on such benchmarks would provide stronger evidence supporting the claim that the enhanced understanding capability of the Encoder successfully translates into improved, higher-level reasoning performance in the Decoder (Generation module).

Reference

[1] Chaorui Deng, Deyao Zhu, Kunchang Li, Chenhui Gou, Feng Li, Zeyu Wang, Shu Zhong, Weihao Yu, Xiaonan Nie, Ziang Song, et al. Emerging properties in unified multimodal pretraining. arXiv preprint arXiv:2505.14683, 2025.

[2] Xiaokang Chen, Zhiyu Wu, Xingchao Liu, Zizheng Pan, Wen Liu, Zhenda Xie, Xingkai Yu, and Chong Ruan. Janus-pro: Unified multimodal understanding and generation with data and model scaling. arXiv preprint arXiv:2501.17811, 2025b.

[3] Jinheng Xie, Weijia Mao, Zechen Bai, David Junhao Zhang, Weihao Wang, Kevin Qinghong Lin, Yuchao Gu, Zhijie Chen, Zhenheng Yang, and Mike Zheng Shou. Show-o: One single transformer to unify multimodal understanding and generation. arXiv preprint arXiv:2408.12528, 2024a.

[4] Yuwei Niu, Munan Ning, Mengren Zheng, Weiyang Jin, Bin Lin, Peng Jin, Jiaqi Liao, Chaoran Feng, Kunpeng Ning, Bin Zhu, Li Yuan. Wise: A world knowledge-informed semantic evaluation for text-to-image generation. arXiv preprint arXiv:2503.07265, 2025.

**Questions:**

please see weakness

---

### Note · Authors · 2025-11-12

I have read and agree with the venue's withdrawal policy on behalf of myself and my co-authors.